# Energy-Guided Continuous Entropic Barycenter Estimation for General Costs

**Alexander Kolesov**
Skolkovo Institute of Science and Technology
Artificial Intelligence Research Institute
Moscow, Russia
a.kolesov@skoltech.ru

**Petr Mokrov & Igor Udovichenko**
Skolkovo Institute of Science and Technology
Moscow, Russia
{p.mokrov, i.udovichenko}@skoltech.ru

**Milena Gazdieva**
Skolkovo Institute of Science and Technology
Artificial Intelligence Research Institute
Moscow, Russia
m.gazdieva@skoltech.ru

**Gudmund Pammer**
Graz University of Technology
Graz, Austria
gudmund.pammer@tugraz.at

**Anastasis Kratsios**
Vector Institute, McMaster University
Ontario, Canada
kratsioa@mcmaster.ca

**Evgeny Burnaev & Alexander Korotin**
Skolkovo Institute of Science and Technology
Artificial Intelligence Research Institute
Moscow, Russia
{e.burnaev, a.korotin}@skoltech.ru

## Abstract

Optimal transport (OT) barycenters are a mathematically grounded way of averaging probability distributions while capturing their geometric properties. In short, the barycenter task is to take the average of a collection of probability distributions w.r.t. given OT discrepancies. We propose a novel algorithm for approximating the continuous Entropic OT (EOT) barycenter for arbitrary OT cost functions. Our approach is built upon the dual reformulation of the EOT problem based on weak OT, which has recently gained the attention of the ML community. Beyond its novelty, our method enjoys several advantageous properties: (i) we establish quality bounds for the recovered solution; (ii) this approach seamlessly interconnects with the Energy-Based Models (EBMs) learning procedure enabling the use of well-tuned algorithms for the problem of interest; (iii) it provides an intuitive optimization scheme avoiding min-max, reinforce and other intricate technical tricks. For validation, we consider several low-dimensional scenarios and image-space setups, including *non-Euclidean* cost functions. Furthermore, we investigate the practical task of learning the barycenter on an image manifold generated by a pretrained generative model, opening up new directions for real-world applications. Our code is available at https://github.com/justkolesov/EnergyGuidedBarycenters.

## 1 Introduction

Averaging is a fundamental concept in mathematics and plays a central role in numerous applications. While it is a straightforward operation when applied to scalars or vectors in a linear space, the situation complicates when working in the space of probability distributions. Here, simple convex combinations can be inadequate or even compromise essential geometric features, which necessitates a different way of taking averages. To address this issue, one may carefully select a measure of distance that properly captures similarity in the space of probabilities. Then, the task is to find a procedure which identifies a 'center' that, on average, is closest to the reference distributions. One good choice for comparing and averaging probability distributions is provided by the family of Optimal Transport (OT) discrepancies [110]. They have clear geometrical meaning and practical

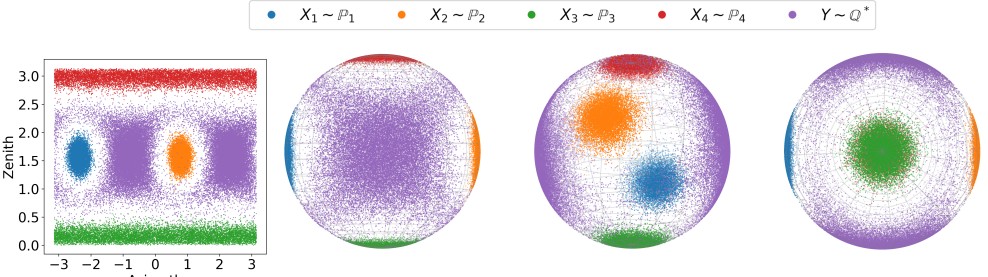

(a) The unfolded sphere.     (b) The sphere viewed from different viewpoints.

Figure 1: Entropic barycenter $\mathbb{Q}^*$ (5) of $N = 4$ von Mises distributions $\mathbb{P}_n$ on the sphere (see §5.1) estimated with our barycenter solver (Algorithm 1). The used transport costs are $c_k(x_k, y) = \frac{1}{2}\left(\arccos\langle x_k, y\rangle\right)^2$.

interpretation [89, 97]. The corresponding problem of averaging probability distributions using OT discrepancies is known as the OT barycenter problem [1]. OT-based barycenters find application in various practical domains: domain adaptation [79], shape interpolation [98], Bayesian inference [101, 102], text scoring [18], style transfer [80], reinforcement learning [77].

Over the past decade, the substantial demand from practitioners sparked the development of various methods tackling the barycenter problem. The research community's initial efforts were focused on the discrete OT barycenter setting, see Appendix B.1 for more details. The **continuous setting** turns out to be even more challenging, with only a handful of recent works devoted to this setup [72, 17, 59, 55, 32, 82, 14]. Most of these works are devoted to specific OT cost functions, e.g., deal with $\ell_2^2$ barycenters [59, 55, 32, 82]; while others require non-trivial *a priori* selections [72] and have limiting expressivity and generative ability [17, 14], see §3 for a detailed discussion.

**Contributions**. We propose a novel approach for solving Entropy-regularized OT (EOT) barycenter problems, which alleviates the aforementioned limitations of existing continuous OT solvers.

1. We reveal an elegant reformulation of the EOT barycenter problem by combining weak dual form of EOT with the congruence condition (§4.1); we derive a simple optimization procedure which closely relates to the standard training algorithm of Energy-Based models (§4.2).
2. We establish the generalization bounds as well as the universal approximation guarantees for our recovered EOT plans, which push the reference distributions to the barycenter (§4.3).
3. We validate the applicability of our approach on various toy and large-scale setups, including the RGB image domain (§5). In contrast to previous works, we also pay attention to non-Euclidean OT costs. Specifically, we conduct a series of experiments looking for a barycenter on an image manifold of a pretrained GAN. In principle, the image manifold support may contribute to the interpretability and plausibility of the resulting barycenter distribution in downstream tasks.

**Notations.** We write $\overline{K} = \{1, 2, \ldots, K\}$. Throughout the paper $\mathcal{X} \subset \mathbb{R}^{D'}, \mathcal{Y} \subset \mathbb{R}^D$ and $\mathcal{X}_k \subset \mathbb{R}^{D_k}$ ($k \in \overline{K}$) are compact subsets of Euclidean spaces. Continuous functions on $\mathcal{X}$ are denoted as $\mathcal{C}(\mathcal{X})$. Probability distributions on $\mathcal{X}$ are $\mathcal{P}(\mathcal{X})$. Absolutely continuous probability distributions on $\mathcal{X}$ are denoted by $\mathcal{P}_{\mathrm{ac}}(\mathcal{X}) \subset \mathcal{P}(\mathcal{X})$. Given $\mathbb{P} \in \mathcal{P}(\mathcal{X}), \mathbb{Q} \in \mathcal{P}(\mathcal{Y})$, we use $\Pi(\mathbb{P}, \mathbb{Q})$ to designate the set of *transport plans*, i.e., probability distributions on $\mathcal{X} \times \mathcal{Y}$ with the first and second marginals given by $\mathbb{P}$ and $\mathbb{Q}$, respectively. The density of $\mathbb{P} \in \mathcal{P}_{\mathrm{ac}}(\mathcal{X})$ w.r.t. the Lebesgue measure is denoted by $\frac{d\mathbb{P}(x)}{dx}$.

## 2 Background

First, we recall the formulations of EOT (§2.1) and the barycenter problem (§2.2). Next, we clarify the computational setup of the considered EOT barycenter task (§2.3).

### 2.1 Entropic Optimal Transport

Consider distributions $\mathbb{P} \in \mathcal{P}_{\mathrm{ac}}(\mathcal{X}), \mathbb{Q} \in \mathcal{P}_{\mathrm{ac}}(\mathcal{Y})$, a continuous cost function $c : \mathcal{X} \times \mathcal{Y} \to \mathbb{R}$ and a regularization parameter $\epsilon > 0$. The *entropic optimal transportation* (EOT) problem between $\mathbb{P}$ and $\mathbb{Q}$ [15, 78] consists of finding a minimizer of

$$\mathrm{EOT}_{c,\epsilon}(\mathbb{P}, \mathbb{Q}) \stackrel{\text{def}}{=} \min_{\pi \in \Pi(\mathbb{P}, \mathbb{Q})} \left\{ \mathop{\mathbb{E}}_{(x,y) \sim \pi} c(x, y) - \epsilon \mathop{\mathbb{E}}_{x \sim \mathbb{P}} H(\pi(\cdot|x)) \right\}. \tag{1}$$

Note that (1) is not the only way to formulate EOT. One more popular and equivalent formulation [19, 83, 36] substitutes the conditional entropy term $\mathbb{E}_{x \sim \mathbb{P}} H(\pi(\cdot|x))$ in (1) with full entropy $H(\pi)$.

A minimizer $\pi^* \in \Pi(\mathbb{P}, \mathbb{Q})$ of (1) is called the EOT plan; its existence and uniqueness are guaranteed, see, e.g., [16, Th. 3.3]. In practice, we usually do not require the EOT plan $\pi^*$ but its conditional distributions $\pi^*(\cdot|x) \in \mathcal{P}(\mathcal{Y})$ as they prescribe how points $x \in \mathcal{X}$ are stochastically mapped to $\mathcal{Y}$ [42, §2]. We refer to $\pi^*(\cdot|x)$ as the *conditional plans*.

**Weak OT dual formulation of the EOT problem**. The EOT problem permits several dual formulations. In our paper, we use the one derived from the weak OT theory [39, Theorem 9.5]:

$$\text{EOT}_{c,\epsilon}(\mathbb{P}, \mathbb{Q}) = \sup_{f \in \mathcal{C}(\mathcal{Y})} \left\{ \mathbb{E}_{x \sim \mathbb{P}} f^C(x) + \mathbb{E}_{y \sim \mathbb{Q}} f(y) \right\}, \tag{2}$$

where $f^C : \mathcal{X} \to \mathbb{R}$ is the so-called **weak** entropic $c$-transform [9, Eq. 1.2] of the function (*potential*) $f$. The transform is defined by

$$f^C(x) \stackrel{\text{def}}{=} \min_{\mu \in \mathcal{P}(\mathcal{Y})} \left\{ \mathbb{E}_{y \sim \mu} c(x, y) - \epsilon H(\mu) - \mathbb{E}_{y \sim \mu} f(y) \right\}. \tag{3}$$

We use the capital $C$ in $f^C$ to distinguish the weak transform from the classic $c$-transform [89, §1.6] or $(c, \epsilon)$-transform [76, §2]. In particular, formulation (2) is not to be confused with the conventional EOT dual, see [78, Appendix A].

For each $x \in \mathcal{X}$, the minimizer $\mu_x^f \in \mathcal{P}(\mathcal{Y})$ of the weak $c$-transform (3) exists and is unique. Its density has particular form [78, Theorem 1]. Let $Z_c(f, x) \stackrel{\text{def}}{=} \int_{\mathcal{Y}} \exp\left(\frac{f(y) - c(x,y)}{\epsilon}\right) dy$, then

$$\frac{d\mu_x^f(y)}{dy} \stackrel{\text{def}}{=} \frac{1}{Z_c(f, x)} \exp\left(\frac{f(y) - c(x, y)}{\epsilon}\right). \tag{4}$$

By substituting (4) into (3) and carrying out straightforward manipulations, we arrive at an explicit formula $f^C(x) = -\epsilon \log Z_c(f, x)$, see [78, Equation (14)].

## 2.2 Entropic OT Barycenter

Consider distributions $\mathbb{P}_k \in \mathcal{P}_{ac}(\mathcal{X}_k)$ and continuous cost functions $c_k(\cdot, \cdot) : \mathcal{X}_k \times \mathcal{Y} \to \mathbb{R}$ for $k \in \overline{K}$. Given weights $\lambda_k > 0$ with $\sum_{k=1}^K \lambda_k = 1$, the EOT Barycenter problem is:

$$\mathcal{L}^* \stackrel{\text{def}}{=} \inf_{\mathbb{Q} \in \mathcal{P}(\mathcal{Y})} \sum_{k=1}^K \lambda_k \text{EOT}_{c_k,\epsilon}(\mathbb{P}_k, \mathbb{Q}), \tag{5}$$

The case where $\epsilon = 0$ corresponds to the classical OT barycenter [1] and falls out of the scope of this paper. Note that the majority of previous research [20, 22, 30, 25, 68, 67] consider a bit different but equivalent EOT barycenter formulation, i.e., which has the **same minimizers**. The objective (5) is known as *Schrödinger* barycenter problem [15, Table 1], see the extended discussion in Appendix B.3. It is worth noting that under mild assumptions the barycenter $\mathbb{Q}^*$ which delivers optimal value of (5) exists and is unique, see Appendix A.7.

## 2.3 Computational aspects of the EOT barycenter task

Barycenter problems, such as (5), are known to be challenging in practice [2]. To our knowledge, even when $\mathbb{P}_1, \ldots, \mathbb{P}_K$ are Gaussian distributions, there is no direct analytical solution neither for our entropic case ($\epsilon > 0$), see the additional discussion in App. C.4, nor for the unregularized case [3, $\epsilon = 0$]. Moreover, in real-world scenarios, the distributions $\mathbb{P}_k$ ($k \in \overline{K}$) are typically not available explicitly but only through empirical samples (datasets). This aspect leads to the next **learning setup**.

We assume that each $\mathbb{P}_k$ is accessible only by a limited number of i.i.d. empirical samples $X_k = \{x_k^1, x_k^2, \ldots x_k^{N_k}\} \sim \mathbb{P}_k$. Our aim is to approximate the optimal conditional plans $\pi_k^*(\cdot|x_k)$ between the entire source distributions $\mathbb{P}_k$ and the entire (unknown) barycenter $\mathbb{Q}^*$ solving (5). The recovered plans should provide the *out-of-sample* estimation, i.e., allow generating samples from $\pi_k^*(\cdot|x_k^{\text{new}})$, where $x_k^{\text{new}}$ is a new sample from $\mathbb{P}_k$ which is not necessarily present in the train sample.

This setup corresponds to **continuous OT** [72, 59]. It differs from the **discrete OT** setup [19, 20] which aims to solve the barycenter task for *discrete* empirical distributions. Discrete OT are not well-suited for the out-of-sample estimation required in the continuous OT setup.

# 3 Related works

The taxonomy of OT solvers is large. Due to space constraints, we discuss here only methods within the *continuous OT learning setup* that solve the (E-)OT barycenter problem. These methods approximate OT maps or plans between the distributions $\mathbb{P}_k$ and the barycenter $\mathbb{Q}^*$ rather than just their empirical counterparts that are available from the training samples. A broader discussion of general-purpose discrete/continuous (E-)OT solvers is in Appendix B.1.

**Continuous OT barycenter solvers** are based on the unregularized or regularized OT barycenter problem within the continuous OT learning setup. The works [59, 32, 82, 55] are designed *exclusively* for the quadratic Euclidean cost $\ell^2(x,y) \stackrel{\text{def}}{=} \frac{1}{2}\|x-y\|_2^2$. The OT problem with this particular cost exhibits several advantageous theoretical properties [4, §2] which are exploited by the aforementioned articles to build efficient pro-

| Method | Admissible OT costs | Learns OT plans | Max considered data dim | Regularization |
|--------|---------------------|-----------------|-------------------------|----------------|
| [72] | general | yes | 8D, no images | Entropic/Quadratic with *fixed* prior |
| [17] | general | no | 1x32x32 (MNIST) | Entropic (Sinkhorn) |
| [59] | only $l_2^2$ | yes | 256D, no images | requires *fixed* prior |
| [32] | only $l_2^2$ | yes | 1x28x28 (MNIST) | no |
| [55] | only $l_2^2$ | yes | 3x64x64 (CelebA, etc.) | no |
| [82] | only $l_2^2$ | yes | 1x28x28 (MNIST) | Entropic |
| [14] | general | yes | 256D, Gaussians only | Entropic/Quadratic |
| **Ours** | general | yes | 3x64x64 (CelebA) | Entropic |

Table 1: Comparison of continuous OT bary solvers

cedures for barycenter estimation algorithms. In particular, [59, 32] utilize ICNNs [6] which parameterize convex functions, and [82] relies on a specific tree-based Schrödinger Bridge reformulation. In contrast, our proposed approach is designed to handle the EOT problem with *arbitrary* cost functions $c_1, \ldots, c_K$. In [72], they also consider regularized OT with non-Euclidean costs. Similar to our method, they take advantage of the dual formulation and exploit the so-called congruence condition (§4). However, their optimization procedure substantially differs. It necessitates selecting a *fixed prior* for the barycenter, which can be non-trivial. The work [14] takes a step further by directly optimizing the barycenter distribution in a variational manner, eliminating the need for a *fixed prior*. This modification increases the complexity of optimization and requires specific parametrization of the variational barycenter. In [17], the authors also parameterize the barycenter as a generative model. Their approach does not recover the OT plans, which differs from our learning setup (§2.3). A summary of the key properties is provided in Table 1, highlighting the fact that our approach overcomes many imperfections of competing methods. We are also aware of the novel continuous OT barycenter solver [52]. This approach is more recent than ours and is *significantly* based on the ideas from our article. Because of this, we exclude it from our comparisons.

## 4 Proposed Barycenter Solver

In the first two subsections, we work out our optimization objective (§4.1) and its practical implementation (§4.2). In §4.3, we alleviate the gap between the theory and practice by offering finite sample approximation guarantees and universality of NNs to approximate the solution.

### 4.1 Deriving the optimization objective

In what follows, we analyze (5) from the dual perspectives. We introduce $\mathcal{L} : \mathcal{C}(\mathcal{Y})^K \to \mathbb{R}$:

$$\mathcal{L}(f_1, \ldots, f_K) \stackrel{\text{def}}{=} \sum_{k=1}^K \lambda_k \left\{ \mathbb{E}_{x_k \sim \mathbb{P}_k} f_k^{C_k}(x_k) \right\} \qquad \left[ \quad = -\epsilon \sum_{k=1}^K \lambda_k \left\{ \mathbb{E}_{x_k \sim \mathbb{P}_k} \log Z_{c_k}(f_k, x_k) \right\} \right].$$

Here $f_k^{C_k}$ denotes the weak entropic $c_k$-transform (3) of $f_k$. Following §2.1, we see that it coincides with $-\epsilon \log Z_{c_k}(f_k, x_k)$. Below we formulate our main theoretical result, which will allow us to solve the EOT barycenter task without optimization over all distributions on $\mathcal{Y}$.

**Theorem 4.1** (Dual formulation of the EOT barycenter problem [proof ref.]). *Problem* (5) *permits the following dual formulation:*

$$\mathcal{L}^* = \sup_{\substack{f_1, \ldots, f_K \in \mathcal{C}(\mathcal{Y}); \\ \sum_{k=1}^K \lambda_k f_k = 0}} \mathcal{L}(f_1, \ldots, f_K). \tag{6}$$

We refer to the constraint $\sum_{k=1}^K \lambda_k f_k = 0$ as the **congruence** condition w.r.t. weights $\{\lambda_k\}_{k=1}^K$. The potentials $f_k$ appearing in (6) play the same role as in (2). Notably, when $\mathcal{L}(f_1, \ldots, f_K)$ is close to $\mathcal{L}^*$, the conditional optimal transport plans $\pi_k^*(\cdot|x_k), x_k \in \mathcal{X}_k$, between $\mathbb{P}_k$ and the barycenter distribution $\mathbb{Q}^*$ can be approximately recovered through the potentials $f_k$. This intuition is formalized in Theorem 4.2 below. First, for $f_k \in \mathcal{C}(\mathcal{Y})$, we define

$$\mathrm{d}\pi^{f_k}(x_k, y) \stackrel{\text{def}}{=} \mathrm{d}\mu_{x_k}^{f_k}(y)\mathrm{d}\mathbb{P}_k(x_k)$$

and set $\mathbb{Q}^{f_k} \in \mathcal{P}(\mathcal{Y})$ to be the second marginal of $\pi^{f_k}$.

**Theorem 4.2** (Quality bound of plans recovered from dual potentials [proof ref.]). *Let* $\{f_k\}_{k=1}^K, f_k \in \mathcal{C}(\mathcal{Y})$ *be congruent potentials. Then we have*

$$\mathcal{L}^* - \mathcal{L}(f_1, \ldots, f_K) = \epsilon \sum_{k=1}^K \lambda_k KL\left(\pi_k^* \| \pi^{f_k}\right) \geq \epsilon \sum_{k=1}^K \lambda_k KL\left(\mathbb{Q}^* \| \mathbb{Q}^{f_k}\right), \tag{7}$$

*where* $\pi_k^* \in \Pi(\mathbb{P}_k, \mathbb{Q}^*), k \in \overline{K}$ *are the EOT plans between* $\mathbb{P}_k$ *and the barycenter distribution* $\mathbb{Q}^*$.

**Algorithm 1:** EOT barycenters via Energy-Based Modelling

---

**Input:** Distributions $\mathbb{P}_k$, $k \in \overline{K}$ accessible by samples; cost functions $c_k(x_k, y) : \mathcal{X}_k \times \mathcal{Y} \to \mathbb{R}$; the regularization coeff. $\epsilon > 0$; barycenter averaging coeff. $\lambda_k > 0 : \sum_{k=1}^{K} \lambda_k = 1$; MCMC procedure `MCMC_proc`; batch size $S > 0$; NNs $f_{\theta,k} : \mathcal{Y} \to \mathbb{R}$, s.t. $\sum_{k=1}^{K} \lambda_k f_{\theta,k} \equiv 0$ (see §4.2).

**Output:** Trained NNs $f_{\theta^*,k}$ recovering the conditional OT plans between $\mathbb{P}_k$ and barycenter $\mathbb{Q}^*$.

> **for** $iter = 1, 2, \dots$ **do**
> > **for** $k = 1, 2, \dots, K$ **do**
> > > Sample batch $\{x_k^s\}_{s=1}^{S} \sim \mathbb{P}_k$;
> > > Draw $Y_k = \{y_k^s\}_{s=1}^{S}$ with MCMC:
> > > > $y_k^s = \texttt{MCMC\_proc}\left( \frac{f_{\theta,k}(\cdot) - c_k(x_k^s, \cdot)}{\epsilon} \right)$;
> > >
> > > $\widehat{L}_k \leftarrow -\lambda_k \frac{1}{S} \left[ \sum_{s=1}^{S} f_{\theta,k}(y_k^s) \right]$;
> >
> > $\widehat{L} \leftarrow \sum_{k=1}^{K} \widehat{L}_k$; Update $\theta$ by using $\frac{\partial \widehat{L}}{\partial \theta}$;

---

According to Theorem 4.2, an approximate solution $\{f_k\}_{k=1}^{K}$ of (6) yields distributions $\pi^{f_k}$ which are close to the optimal plans $\pi_k^*$. Each $\pi^{f_k}$ is formed by conditional distributions $\mu_{x_k}^{f_k}$, c.f. (4), with closed-form energy function, i.e., the unnormalized log-likelihood. Consequently, one can generate samples from $\mu_{x_k}^{f_k}$ using standard MCMC techniques [7]. In the next subsection, we stick to the practical aspects of optimization of (6), which bears certain similarities to the training of Energy-Based models [69, 100, EBM].

**Relation to prior works.** Works [72, 59] also aim to first get the dual potentials and then recover the barycenter, see the discussion in §3 for more details.

### 4.2 Practical Optimization Algorithm

To maximize the dual EOT barycenter objective (6), we replace the potentials $f_k \in \mathcal{C}(\mathcal{Y})$ for $k \in \overline{K}$ with neural networks $f_{\theta,k}$, $\theta \in \Theta$. In order to eliminate the constraint in (6), we parametrize $f_{\theta,k}$ as $g_{\theta_k} - \sum_{k'=1}^{K} \lambda_{k'} g_{\theta_{k'}}$, where $\{g_{\theta_k} : \mathbb{R}^D \to \mathbb{R}, \theta_k \in \Theta_k\}_{k=1}^{K}$ are neural networks. This parameterization automatically ensures the congruence condition $\sum_{k=1}^{K} \lambda_k f_{\theta,k} \equiv 0$. Note that $\Theta = \Theta_1 \times \cdots \times \Theta_K$ and $\theta = (\theta_1, \dots, \theta_K) \in \Theta$. Our objective function for (6) is

$$L(\theta) \overset{\text{def}}{=} -\epsilon \sum_{k=1}^{K} \lambda_k \left\{ \underset{x_k \sim \mathbb{P}_k}{\mathbb{E}} \log Z_{c_k}(f_{\theta,k}, x_k) \right\}. \tag{8}$$

The direct computation of the normalizing constant $Z_{c_k}$ may be infeasible. Still, the gradient of $L$ with respect to $\theta$ can be derived similarly to [78, Theorem 3]:

**Theorem 4.3** (Gradient of the dual EOT barycenter objective [proof ref]). *The gradient of $L$ satisfies*

$$\frac{\partial}{\partial \theta} L(\theta) = -\sum_{k=1}^{K} \lambda_k \underset{x_k \sim \mathbb{P}_k}{\mathbb{E}} \left\{ \underset{y \sim \mu_{x_k}^{f_{\theta,k}}}{\mathbb{E}} \left[ \frac{\partial}{\partial \theta} f_{\theta,k}(y) \right] \right\}. \tag{9}$$

With this result, we can describe our proposed algorithm which maximizes $L$ using (9).

TRAINING. To perform stochastic gradient ascent step w.r.t. $\theta$, we approximate (9) with Monte-Carlo by drawing samples from $d\pi^{f_{\theta,k}}(x_k, y) = d\mu_{x_k}^{f_{\theta,k}}(y) d\mathbb{P}_k(x_k)$. Analogously to [78, §3.2], this can be achieved by a simple two-stage procedure. At first, we draw a random vector $x_k$ from $\mathbb{P}_k$. This is done by picking a random empirical sample from the available dataset $X_k$. Then, we need to draw a sample from the distribution $\mu_{x_k}^{f_{\theta,k}}$. Since we know the negative **energy** (unnormalized log density) of $\mu_{x_k}^{f_{\theta,k}}$ by (4), we can sample from this distribution by applying an MCMC procedure which uses the negative energy function $\epsilon^{-1}(f_{\theta,k}(y) - c_k(x_k, y))$ as the input. Our findings are summarized in Algorithm 1. Note that typical MCMC needs the energy functions, in particular, costs $c_k$, to be differentiable. Otherwise, one can consider *gradient-free* procedures, e.g., [92, 104].

In all our experiments, we use ULA [86, §1.4.1] as a `MCMC_proc`. It is a conventional MCMC algorithm. Specifically, in order to draw a sample $y_k \sim \mu_{x_k}^{f_{\theta,k}}$, where $x_k \in \mathcal{X}_k$, we initialize $y_k^{(0)}$ from the $D$−dimensional distribution $\mathcal{N}(0, I_D)$ and then iterate the discretized Langevin dynamics:

$$y_k^{(l+1)} \leftarrow y_k^{(l)} + \frac{\eta}{2\epsilon} \nabla_y \big( f_{\theta,k}(y) - c(x_k, y) \big) \Big|_{y=y_k^{(l)}} + \sqrt{\eta} \xi_l \,,$$

where $\xi_l \sim \mathcal{N}(0, I_D)$, $l \in \{0, 1, 2, \ldots, L\}$, $L$ is a number of steps, and $\eta > 0$ is a step size. Note that the iteration procedure above could be straightforwardly adapted to a batch scenario, i.e., we can simultaneously simulate the whole batch of samples $Y_k^{(l)}$ conditioned on $X_k^{(l)}$. The particular values of number of steps $L$ and step size $\eta$ are reported in the details of the experiments, see Appendix C. An alternative importance sampling-based approach for optimizing (9) is presented in Appendix D.

INFERENCE. We use the same ULA procedure for sampling from the recovered optimal conditional plans $\pi^{f_{\theta^*}, k}(\cdot | x_k)$, see the details on the hyperparameters $L, \eta$ in §5.

**Relation to prior works.** Learning a distribution of interest via its energy function (EBMs) is a well-established direction in generative modelling research [69, 111, 28, 100]. Similar to ours, the key step in most energy-based approaches is the MCMC procedure which recovers samples from a distribution accessible only by an unnormalized log density. Typically, various techniques are employed to improve the stability and convergence speed of MCMC, see, e.g., [27, 34, 113]. The majority of these techniques can be readily adapted to complement our approach. At the same time, the primary goal of this study is to introduce and validate the methodology for computing EOT barycenters in an energy-guided manner. Therefore, we opt for the **simplest** MCMC algorithm, even **without the replay buffer** [46], as it serves our current objectives.

## 4.3   Generalization Bounds and Universal Approximation with Neural Nets

In this subsection, we answer the question of how far the recovered plans are from the EOT plan $\pi_k^*$ between $\mathbb{P}_k$ and $\mathbb{Q}$. In practice, for each distribution $\mathbb{P}_k$ we know only the empirical samples $X_k = \{x_k^1, x_k^2, \ldots x_k^{N_k}\} \sim \mathbb{P}_k$, i.e., finite datasets. Besides, the available potentials $f_k$, $k \in \overline{K}$ come from restricted classes of functions and satisfy the congruence condition. More precisely, we have $f_k = g_k - \sum_{k=1}^K \lambda_k g_k$ (§4.2), where each $g_k$ is picked from some class $\mathcal{G}_k$ of neural networks. Formally, we write $(f_1, \ldots, f_K) \in \overline{\mathcal{F}}$ to denote the congruent potentials constructed this way from the functional classes $\mathcal{G}_1, \ldots, \mathcal{G}_K$. Hence, in practice, we optimize the *empirical version* of (8):

$$\max_{(f_1, \ldots, f_K) \in \overline{\mathcal{F}}} \widehat{\mathcal{L}}(f_1, \ldots, f_K) \stackrel{\text{def}}{=} \max_{(f_1, \ldots, f_K) \in \overline{\mathcal{F}}} \sum_{k=1}^K \frac{\lambda_k}{N_k} \sum_{n=1}^{N_k} f_k^{C_k}(x_k^n);$$

$$(\widehat{f_1}, \ldots \widehat{f_K}) \stackrel{\text{def}}{=} \arg\max_{(f_1, \ldots, f_K) \in \overline{\mathcal{F}}} \widehat{\mathcal{L}}(f_1, \ldots, f_k).$$

A natural question arises: ***How close are the recovered plans*** $\pi^{\widehat{f_k}}$ ***to the EOT plans*** $\pi_k^*$ between $\mathbb{P}_k$ and $\mathbb{Q}^*$? Since our objective (8) is a sum of integrals over distributions $\mathbb{P}_k$, the generalization error can be straightforwardly decomposed into the estimation and approximation parts.

**Proposition 4.4** (Decomposition of the generalization error [proof ref.]). *The following bound holds:*

$$\epsilon \mathbb{E} \sum_{k=1}^K \lambda_k KL\left(\pi_k^* \| \pi^{\widehat{f_k}}\right) \leq \overbrace{2 \sum_{k=1}^K \lambda_k \mathbb{E} Rep_{X_k}(\mathcal{F}_k^{C_k}, \mathbb{P}_k)}^{\text{Estimation error (upper bound)}} + \overbrace{\left[\mathcal{L}^* - \max_{(f_1, \ldots, f_K) \in \overline{\mathcal{F}}} \mathcal{L}(f_1, \ldots, f_K)\right]}^{\text{Approximation error}}, \quad (10)$$

*where $\mathcal{F}_k^{C_k} \stackrel{\text{def}}{=} \{f_k^{C_k} \mid (f_1, \ldots, f_K) \in \overline{\mathcal{F}}\}$, and the expectations are taken w.r.t. the random realization of the datasets $X_1 \sim \mathbb{P}_1, \ldots, X_K \sim \mathbb{P}_K$. Here $Rep_{X_k}(\mathcal{F}_k^{C_k}, \mathbb{P}_k)$ is the standard notion of the* ***representativeness*** *of the sample $X_k$ w.r.t. functional class $\mathcal{F}_k^{C_k}$ and distribution $\mathbb{P}_k$, see App. A.5.*

To bound the estimation error, we need to further bound the *expected* representativeness $\mathbb{E} Rep_{X_k}(\mathcal{F}_k^{C_k}, \mathbb{P}_k)$. Doing preliminary analysis, we found that it does not depend that much on the complexity of the functional class $\mathcal{F}_k$. However, it seems to heavily depend on the properties of the cost $c_k$. We derive two bounds: a general one for Lipschitz (in $x$) costs and a better one for the feature-based quadratic costs.

**Theorem 4.5** (Bound on $\mathbb{E} Rep$ w.r.t. $C_k$-transform classes [proof ref.]). *(a) Let $\mathcal{F}_k \subset \mathcal{C}(\mathcal{Y})$. Assume that $c_k(x, y)$ is Lipschitz in $x$ with the same Lipschitz constant for all $y \in \mathcal{Y}$. Then*

$$\mathbb{E} Rep_{X_k}(\mathcal{F}_k^{C_k}, \mathbb{P}_k) \leq O\left(N_k^{-1/(D_k+1)}\right). \quad (11)$$

*(b) Let $c_k(x_k, y) = \frac{1}{2}\|u_k(x_k) - v(y)\|^2$, $\mathcal{F}_k$ be a **bounded** (w.r.t. the supremum norm) subset of $\mathcal{C}(\mathcal{Y})$, $u_k : \mathcal{X}_k \to \mathbb{R}^{D''}$ and $v : \mathcal{Y} \to \mathbb{R}^{D''}$ be continuous functions. Then*

$$\mathbb{E} Rep_{X_k}(\mathcal{F}_k^{C_k}, \mathbb{P}_k) \leq O\left(N_k^{-1/2}\right). \quad (12)$$

Substituting (11) or (12) to (10) immediately provides the *statistical consistency* when $N_1, \ldots, N_K \to \infty$, i.e., vanishing of the estimation error when the sample size grows.

The case **(a)** here is not very practically useful as the rate suffers from the curse of dimensionality. Still, this result points to one intriguing property of our solver. Namely, we may take **arbitrarily large** set $\mathcal{F}_k$ (even $\mathcal{F}_k = \mathcal{C}(\mathcal{Y})$!) and still have the guarantees of learning the barycenter. This happens because of $C_k$-transforms: informally, they make functions $f_k \in \mathcal{F}_k$ smoother and "simplify" the set $\mathcal{F}_k$. In our experiments, we always work with the costs as in **(b)**. As a result, our estimation error is $O(\sum_{k=1}^K N_k^{-1/2})$; this is a *standard fast and dimension-free convergence rate*. In practice, $\mathcal{F}_k$ are usually neural nets. They are indeed bounded, as required in **(b)**, if their weights are constrained.

While the estimation error usually decreases when the sample sizes tend to infinity, it is natural to wonder whether the approximation error can be also made arbitrarily small. We positively answer this question when the standard fully-connected neural nets (multi-layer perceptrons) are used.

**Theorem 4.6** (Vanishing Approximation Error [proof ref.]). *Let $\sigma : \mathbb{R} \to \mathbb{R}$ be an activation function. Assume that it is non-affine and there is an $\widetilde{x} \in \mathbb{R}$ at which $\sigma$ is differentiable and $\sigma'(\widetilde{x}) \neq 0$. Then for every $\delta > 0$ there exist $K$ multi-layer perceptrons $g_k : \mathbb{R}^D \to \mathbb{R}$ with activations $\sigma$ for which the congruent functions $f_k = g_k - \sum_{k=1}^K \lambda_k g_k$ satisfy*

$$\sum_{k=1}^K \lambda_k KL\left(\pi_k^* \| \pi^{f_k}\right) = (\mathcal{L}^* - \mathcal{L}(f_1, \ldots, f_K))/\epsilon < \delta/\epsilon.$$

*Furthermore, each $g_k$ has width at-most $D + 4$.*

Importantly, our Theorem 4.6 is more than just result on universal approximation since it deals with (i) *congruent* potentials and (ii) entropic $C_k$-transforms. In particular, only specific properties of the entropic $C_k$-transforms allow deriving the desired universal approximation statement, see the proof.

**Summary.** Our results of this section show that both the estimation and approximation errors can be made arbitrarily small given a sufficient amount of data and large neural nets, allowing to perfectly recover the EOT plans $\pi_k^*$.

**Relation to prior works.** To our knowledge, the generalization and the universal approximation are novel results with no analogues established for any other continuous barycenter solver. Our analysis shows that the EOT barycenter objective (8) is well-suited for statistical learning and approximation theory tools. This aspect distinguishes our work from the predecessors, where complex optimization objectives may not be as amenable to rigorous study.

## 4.4 Learning EOT barycenter on data manifold

Averaging complex data distributions by means of EOT barycenter directly in the data space may be undesirable. In particular, for image data domain:

- the entropic barycenter contains noisy images, see, e.g., our MNIST 0/1 experiment, §5.2. This is due to the "blurring bias" bias [15, 49] of our entropic barycenter setup and reliance on MCMC.

- searching for (entropic) barycenter is not very practical for standard OT cost functions like $\ell^2$. It is known that the true unregularized ($\epsilon = 0$) $\ell^2$-barycenter of several image domains consists of just some pixel-wise averages of images from these source domains, which is not practically useful.

To alleviate the problem, we propose solving the (entropic) barycenter problem on some *a priori* known data manifold $\mathcal{M}$, where we want the barycenter to be concentrated on. In our experiments (§5.2, §5.3) these manifolds are given by pre-trained StyleGAN [50] generator models $G : \mathcal{Z} \to \mathcal{Y}$; $\mathcal{Z}$ is the *latent* space, $\mathcal{M} = G(\mathcal{Z})$. Technically speaking, to adapt our Alogithm 1 for manifold-constrained setup, we propose solving the barycenter problem in *latent* space $\mathcal{Z}$ with *modified* cost functions $c_{k,G}(x_k, z) := c_k(x_k, G(z))$. We emphasize that such costs are **general** (not $\ell^2$ cost!) because $G$ is a non-trivial StyleGAN generator. Hence, while our proposed manifold-constrained barycenter learning setup could be used on par with other OT barycenter solvers, these barycenter solvers **should** support general costs. In particular, the majority of competitive methods from Table 1 *are not adjustable to the manifold setup* as they work exclusively with $\ell^2$.

**Relation to prior works.** While the utilization of data manifolds given by pre-trained (foundational) models is ubiquitous in generative modeling, the adaptation of this technique for Optimal Transport barycenter is a novel idea. Apart from our work, this idea is exploited in follow-up paper [52].

# 5 Experiments

We assess the performance of our barycenter solver on small-dimensional illustrative setups (§5.1) and in image spaces (§5.2, §5.3). The source code for our solver is written in the `PyTorch` framework and available at `https://github.com/justkolesov/EnergyGuidedBarycenters`. The experiments are issued in the form of convenient `*.ipynb` notebooks. Reproducing the most challenging experiments (§5.2, §5.3) requires less than 12 hours on a single TeslaV100 GPU. The details of the experiments, extended experimental results are in Appendix C, additional experiments with single-cell data are given in Appendix C.5.

**Disclaimer.** Evaluating how well our solver recovers the EOT barycenter is challenging because the ground truth barycenter is typically unknown. In some cases, the true *unregularized* barycenter ($\epsilon = 0$) can be derived (see below). The EOT barycenter for sufficiently small $\epsilon > 0$ is expected to be close to the unregularized one. Therefore, in most cases, our evaluation strategy is to compare the computed EOT barycenter (for small $\epsilon$) with the unregularized one. In particular, we use this strategy to quantitatively evaluate our solver in the Gaussian case, see Appendix C.4.

## 5.1 Barycenters of Toy Distributions

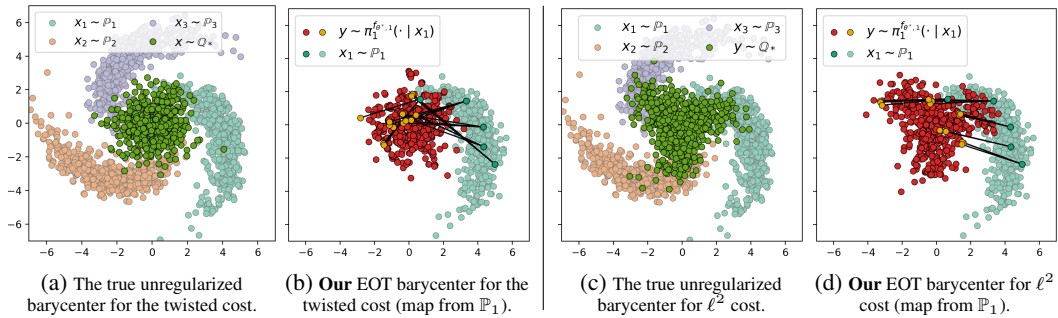

(a) The true unregularized barycenter for the twisted cost.

(b) **Our** EOT barycenter for the twisted cost (map from $\mathbb{P}_1$).

(c) The true unregularized barycenter for $\ell^2$ cost.

(d) **Our** EOT barycenter for $\ell^2$ cost (map from $\mathbb{P}_1$).

Figure 2: *2D twister example*: The true barycenter of 3 comets vs. the one computed by our solver with $\epsilon = 10^{-2}$. Two costs $c_k$ are considered: the twisted cost (2a, 2b) and $\ell^2$ (2c, 2d).

**2D Twister**. Consider the map $u : \mathbb{R}^2 \to \mathbb{R}^2$ which, in the *polar coordinate system*, is represented by $\mathbb{R}_+ \times [0, 2\pi) \ni (r, \theta) \mapsto (r, (\theta - r) \bmod 2\pi)$. The cartesian version of $u$ is presented in Appendix C.1. Let $\mathbb{P}_1, \mathbb{P}_2, \mathbb{P}_3$ be 2-dimensional distributions as shown in Fig. 2a. For these distributions and uniform weights $\lambda_k = \frac{1}{3}$, the unregularized barycenter ($\epsilon = 0$) for the **twisted** cost $c_k(x_k, y) = \frac{1}{2}\|u(x_k) - u(y)\|^2$ can be derived analytically, see Appendix C.1. The barycenter is the centered Gaussian distribution which is also shown in Fig. 2a. We run the experiment for this cost with $\epsilon = 10^{-2}$, and the results are recorded in Fig. 2b. We see that it qualitatively coincides with the true barycenter. For completeness, we also show the EOT barycenter computed with our solver for $\ell^2(x, y) = \frac{1}{2}\|x - y\|^2$ costs (Fig. 2c) and the same regularization $\epsilon$. The true $\ell^2$ barycenter is estimated by using the `free_support_barycenter` solver from POT package [33]. We stress that the twisted cost barycenter and $\ell^2$ barycenter differ, and so do the learned conditional plans: the $\ell^2$ EOT plan (Fig. 2d) expectedly looks more well-structured while for the twisted cost (Fig. 2b) it becomes more chaotic due to non-trivial structure of this cost.

**Sphere.** In this experiment, we look for the barycenter of four von Mises distributions $\mathbb{P}_n$ supported on 3D sphere, see Figure 1. The cost functions are $c_k(x_k, y) = \frac{1}{2}\arccos^2\langle x_k, y\rangle$, the regularization is $\epsilon = 10^{-2}$. The learned potentials $f_{\theta,k}$ operate with ambient $\mathbb{R}^3$ vectors. When performing MCMC, we project each Langevin step to the sphere. Our qualitative results are shown on Figure 1. While the ground truth solution to the considered problem is unknown, the learned barycenter looks reasonable. This showcases the applicability of our approach to non-standard non-quadratic experimental setups.

## 5.2 Barycenters of MNIST Classes 0 and 1

A classic experiment considered in the continuous barycenter literature [32, 55, 82, 17] is averaging of distributions of MNIST 0/1 digits with

Figure 3: Samples from the StyleGAN $G$ defining the polluted manifold $\mathcal{M}$.

weights $(\frac{1}{2}, \frac{1}{2})$ in the grayscale image space $\mathcal{X}_1 = \mathcal{X}_2 = \mathcal{Y} = [-1, 1]^{32 \times 32}$. The true unregularized ($\epsilon = 0$) $\ell^2$-barycenter images $y$ are direct pixel-wise averages $\frac{x_1 + x_2}{2}$ of pairs of images $x_1$ and $x_2$

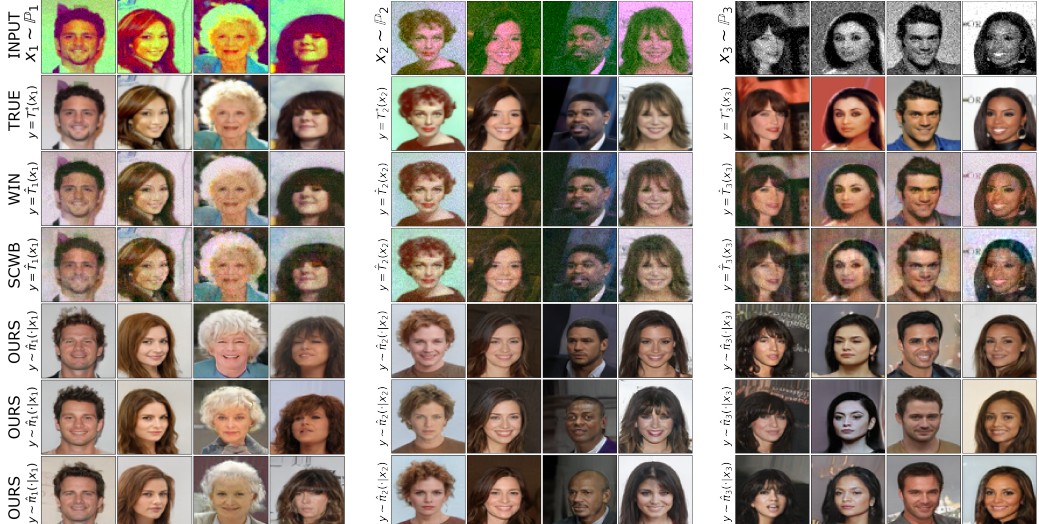

(a) Maps from $\mathbb{P}_1$ to the barycenter.  (b) Maps from $\mathbb{P}_2$ to the barycenter.  (c) Maps from $\mathbb{P}_3$ to the barycenter.

Figure 4: *Experiment on the Ave, celeba! barycenter dataset.* The plots compare the transported inputs $x_k \sim \mathbb{P}_k$ to the barycenter learned by various solvers. The true unregularized $\ell^2$ barycenter of $\mathbb{P}_1, \mathbb{P}_2, \mathbb{P}_3$ are the clean celebrity faces, see [55, §5].

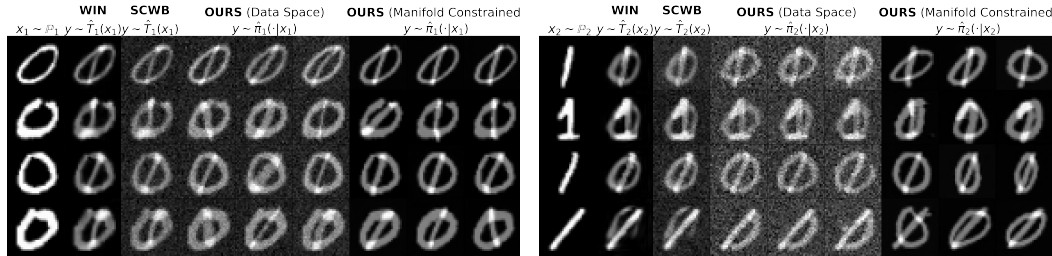

(a) Learned plans from $\mathbb{P}_1$ (zeros) to the barycenter.  (b) Learned plans from $\mathbb{P}_2$ (ones) to the barycenter.

Figure 5: Qualitative comparison of barycenters of MNIST 0/1 digit classes computed with barycenter solvers in the image space w.r.t. the pixel-wise $\ell^2$. Solvers SCWB and WIN only learn the unregularized barycenter ($\epsilon = 0$) directly in the data space. In turn, our solver learns the EOT barycenter in data space as well as it can learn EOT barycenter restricted to the StyleGAN manifold ($\epsilon = 10^{-2}$).

coming from the $\ell^2$ OT plan between 0's ($\mathbb{P}_1$) and 1's ($\mathbb{P}_2$). In Fig. 5, we show the unregularized $\ell^2$ barycenter computed by [32, SCWB], [55, WIN].

**Data space EOT barycenter.** To begin with, we employ our solver to compute the $\epsilon$-regularized EOT $\ell^2$-barycenter directly in the image space $\mathcal{Y}$ for $\epsilon = 10^{-2}$. We emphasize that the true entropic barycenter slightly differs from the unregularized one. To be precise, it is expected that regularized barycenter images are close to the unregularized barycenter images but with additional noise. In Fig. 5, we see that our solver (data space) recovers the noisy barycenter images exactly as expected.

**Manifold-constrained EOT barycenter.** Following the reasoning from §4.4, we propose to restrict the search space for our algorithm to some pre-defined manifold $\mathcal{M}$. As discussed earlier, the support of the image-space unregularized $\ell^2$-barycenter is a certain *subset* of $\mathcal{M}' \stackrel{\text{def}}{=} \{ \frac{x_1 + x_2}{2} \mid x_1 \in \text{Supp}(\mathbb{P}_1), x_2 \in \text{Supp}(\mathbb{P}_2) \}$. To achieve this, we train a StyleGAN [50] model $G : \mathcal{Z} \to \mathcal{Y}$ with $\mathcal{Z} = \mathbb{R}^{512}$ to generate some **even larger** manifold $\mathcal{M} = G(\mathcal{Z})$ which is expected to contain $\mathcal{M}'$. Namely, we use all possible pixel-wise half-sums $\frac{x_1 + x_2}{2}$ of digits 0 as $x_1$ and $\{1, 4, 7\}$ as $x_2$, see Figure 3 with the trained StyleGAN samples. That is, our final constructed manifold $\mathcal{M}$ is **polluted** with additional samples (e.g., averages of digits 0 and 7) which should not to lie in the support of the barycenter. Then, we use our solver with $\epsilon = 10^{-2}$ to search for the barycenter of 0/1 digit distributions on $\mathcal{X}_1, \mathcal{X}_2$ which lies in the latent space $\mathcal{Z}$ w.r.t. costs $c_{k,G}(x, z) \stackrel{\text{def}}{=} \frac{1}{2} \|x - G(z)\|^2$. This can be interpreted as learning the EOT $\ell^2$-barycenter in the ambient space but constrained to

the StyleGAN-parameterized manifold $G(\mathcal{Z})$. The barycenter $\mathbb{Q}^*$ is some distribution of the latent variables $z$, which can be pushed to the manifold $G(\mathcal{Z}) \subset \mathcal{Y}$ via $G(z)$.

The results are in Fig. 5. There is **(a)** no noise compared to the data-space EOT barycenter because of the manifold constraint, and **(b)** our solvers correctly ignores polluted samples from $\mathcal{M}$.

### 5.3 Evaluation on the Ave, celeba! Dataset

In [55], the authors developed a theoretically grounded methodology for finding probability distributions whose un-regularized $\ell^2$ barycenter is known by construction. Based on the CelebA faces dataset [73], they constructed an Ave, celeba! dataset containing 3 degraded subsets of faces. The true $\ell^2$ barycenter w.r.t. the weights $(\frac{1}{4}, \frac{1}{2}, \frac{1}{4})$ is the distribution of Celeba faces itself. This dataset is used to test how well our approach recovers the barycenter.

| *Solver* | $FID\downarrow$ of plans to barycenter | | |
|---|---|---|---|
| | $k=1$ | $k=2$ | $k=3$ |
| SCWB [32] | 56.7 | 53.2 | 58.8 |
| WIN [55] | 49.3 | 46.9 | 61.5 |
| **Ours** | **8.4** (.3) | **8.7** (.3) | **10.2** (.7) |

Table 2: FID scores of images mapped from inputs $\mathbb{P}_k$ to the barycenter.

We follow the EOT manifold-constrained setup (§4.4) and train the StyleGAN on unperturbed celeba faces. This might sound a little bit unfair, but our goal is to demonstrate the learned transport plan to the constrained barycenter rather than unconditional barycenter samples (recall the setup in §2.3). Hence, we learn the constrained EOT barycenter with $\epsilon = 10^{-4}$. In Fig. 4, we present the results, depicting samples from the learned plans from each $\mathbb{P}_k$ to the barycenter. Overall, the map is qualitatively good, although sometimes failures in preserving the image content may occur. This is presumably due to MCMC inference getting stuck in local minima of the energy landscape. For comparison, we also show the results of the solvers by [32, SCWB], [55, WIN]. Additionally, we report the FID score [45] for images mapped to the barycenter in Table 2 (std. deviations for our method correspond to running the inference with different random seeds). Owing to the manifold-constrained setup, the FID score of our solver is significantly smaller.

## 6 Potential Impact, Limitations and Broader Impact

**Potential impact.** In our work, we propose a novel approach for solving EOT barycenter problems which is applicable to *general OT costs*. From the practical viewpoint, we demonstrate the ability to restrict the sought-for barycenter to the *image manifold* by utilizing a pretrained generative model. Our findings may be applicable to a list of important real-world applications, see Appendix B.2. We believe that our large-scale barycenter solver will leverage industrial & socially-important problems.

**Methodological limitations**. The methodological limitations of our approach are mostly the same as those of EBMs. It is worth mentioning the usage of MCMC during the training/inference. The basic ULA algorithm which we use in §4.2 may poorly converge to the desired distribution $\mu_x^f$. In addition, MCMC sampling is time-consuming. We leave the search for more efficient sampling procedures for our solver, e.g., [71, 99, 43, 81, 47, 108, 66, 26], for future research. We also note that our theoretical analysis in §4.3 does not take into the account the optimization errors appearing due to the gradient descent and MCMC. The analysis of these quantities is a completely different domain in machine learning and out of the scope of our work. As the most generative modelling research, we do not attempt to analyse these errors.

**Problem setup limitations.** Our paper aims at solving Entropic OT barycenter problem. In the image data space, due to utilization of the Entropy, the learned barycenter distribution may contain noisy images. However, the utilization of our proposed StyleGAN-inspired manifold technique **entirely** alleviates the problem with the noise. This is demonstrated by our latent-space experiments with MNIST 0/1 (manifold space) and Ave Celeba! dataset.

**Broader impact.** This paper presents work whose goal is to advance the field of Machine Learning. There are many potential societal consequences of our work, none which we feel must be specifically highlighted here.

## 7 Acknowledgements

Skoltech was supported by the Analytical center under the RF Government (subsidy agreement 000000D730321P5Q0002, Grant No. 70-2021-00145 02.11.2021). AK acknowledges financial support from the NSERC Discovery Grant No. RGPIN-2023-04482. We would like to express special thanks to Vladimir Vanovskiy from Skoltech for the insightful discussions and details on geological modelling (Appendix B.2).

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

# A  Proofs

## A.1  Auxiliary Statements

We start by showing some basic properties of the $C$-transform which will be used in the main proofs.

**Proposition A.1** (Properties of the $C$-transform)**.** *Let $f_1, f_2 \colon \mathcal{Y} \to \mathbb{R}$ be two measurable functions which are bounded from below. It holds that*

(i) **Monotonicity***: $f_1 \leq f_2$ implies $f_1^C \geq f_2^C$;*

(ii) **Constant additivity***: $(f_1 + a)^C = f_1^C - a$ for all $a \in \mathbb{R}$;*

(iii) **Concavity***: $(\lambda f_1 + (1 - \lambda) f_2)^C \geq \lambda f_1^C + (1 - \lambda) f_2^C$ for all $\lambda \in [0, 1]$;*

(iv) **Continuity***: $f_1, f_2$ bounded implies $\sup_{x \in \mathcal{X}} |f_1^C(x) - f_2^C(x)| \leq \sup_{y \in \mathcal{Y}} |f_1(y) - f_2(y)|$.*

*Proof of Proposition A.1.* We recall the definition of the $C$-transform

$$f_1^C(x) = \inf_{\mu \in \mathcal{P}(\mathcal{Y})} \left\{ C(x, \mu) - \int_{\mathcal{Y}} f_1(y) \mathrm{d}\mu(y) \right\},$$

where $C(x, \mu) \stackrel{\text{def}}{=} \int_{\mathcal{Y}} c(x, y) \mathrm{d}\mu(y) - \epsilon H(\mu)$. Monotonicity (i) and constant additivity (ii) are immediate from the definition.

To see (iii), observe that the dependence of $\int_{\mathcal{Y}} f_1(y) \mathrm{d}\mu(y)$ on $f_1$ is linear. Thus, $f_1^C$ is the pointwise infimum of a family of linear functionals and thus concave.

Finally, to show (iv) we have by monotonicity of the integral that

$$\left| \int_{\mathcal{Y}} f_1(y) \mathrm{d}\mu(y) - \int_{\mathcal{Y}} f_2(y) \mathrm{d}\mu(y) \right| \leq \sup_{y \in \mathcal{Y}} |f_1(y) - f_2(y)| \tag{13}$$

for any $\mu \in \mathcal{P}(\mathcal{Y})$. For fixed $x \in \mathcal{X}$ we have

$$f_1^C(x) - f_2^C(x) = \inf_{\mu \in \mathcal{P}(\mathcal{Y})} \left[ C(x, \tilde{\mu}) - \int_{\mathcal{Y}} f_1(y) \mathrm{d}\tilde{\mu}(y) \right] - \inf_{\mu \in \mathcal{P}(\mathcal{Y})} \left[ C(x, \mu) - \int_{\mathcal{Y}} f_2(y) \mathrm{d}\mu(y) \right]$$

$$= \sup_{\mu \in \mathcal{P}(\mathcal{Y})} \inf_{\tilde{\mu} \in \mathcal{P}(\mathcal{Y})} \left[ C(x, \tilde{\mu}) - C(x, \mu) - \int_{\mathcal{Y}} f_1(y) \mathrm{d}\tilde{\mu}(y) + \int_{\mathcal{Y}} f_2(y) \mathrm{d}\mu(y) \right].$$

By setting $\tilde{\mu} = \mu$ we increase the value and obtain

$$f_1^C(x) - f_2^C(x) \leq \sup_{\mu \in \mathcal{P}(\mathcal{Y})} \int_{\mathcal{Y}} [f_2(y) - f_1(y)] \, \mathrm{d}\mu(y) \leq \sup_{y \in \mathcal{Y}} |f_1(y) - f_2(y)|, \tag{14}$$

where the last inequality follows from (13). For symmetry reasons, we can swap the roles of $f_1$ and $f_2$ in (14), which yields the claim. $\qquad\square$

## A.2  Proof of Theorem 4.1

*Proof.* By substituting in (5) the primal EOT problems (1) with their dual counterparts (2), we obtain a dual formulation, which is the starting point of our analysis:

$$\mathcal{L}^* = \min_{\mathbb{Q} \in \mathcal{P}(\mathcal{Y})} \sup_{f_1, \dots, f_K \in \mathcal{C}(\mathcal{Y})} \underbrace{\sum_{k=1}^{K} \lambda_k \left\{ \int_{\mathcal{X}_k} f_k^{C_k}(x_k) \mathrm{d}\mathbb{P}_k(x_k) + \int_{\mathcal{Y}} f_k(y) \mathrm{d}\mathbb{Q}(y) \right\}}_{\stackrel{\text{def}}{=} \widetilde{\mathcal{L}}\left(\mathbb{Q}, \{f_k\}_{k=1}^K\right)}. \tag{15}$$

Here, we replaced inf with min because of the existence of the barycenter (§2.2). Moreover, we refer to the entire expression under the min sup as a functional $\widetilde{\mathcal{L}} \colon \mathcal{P}(\mathcal{Y}) \times \mathcal{C}(\mathcal{Y})^K \to \mathbb{R}$. For brevity, we introduce, for $(f_1, \dots, f_K) \in \mathcal{C}(\mathcal{Y})^K$, the notation

$$\bar{f} \stackrel{\text{def}}{=} \sum_{k=1}^{K} \lambda_k f_k \quad \text{and} \quad M \stackrel{\text{def}}{=} \inf_{y \in \mathcal{Y}} \bar{f}(y) = \inf_{\mathbb{Q} \in \mathcal{P}(\mathcal{Y})} \int \bar{f}(y) \mathrm{d}\mathbb{Q}(y), \tag{16}$$

where the equality follows from two elementary observations that **(a)** $M \leq \int \bar{f}(y) \mathrm{d}\mathbb{Q}(y)$ for any $\mathbb{Q} \in \mathcal{P}(\mathcal{Y})$ and **(b)** $\bar{f}(y) = \int \bar{f}(y') \mathrm{d}\delta_y(y')$ where $\delta_y$ denotes a Dirac mass at $y \in \mathcal{Y}$.

On the one hand, $\mathcal{P}(\mathcal{Y})$ is compact w.r.t. the weak topology because $\mathcal{Y}$ is compact, and for fixed potentials $(f_1, \ldots, f_K) \in \mathcal{P}(\mathcal{Y})^K$ we have that $\widetilde{\mathcal{L}}(\cdot, (f_k)_{k=1}^K)$ is continuous and linear. In particular, $\widetilde{\mathcal{L}}(\cdot, (f_k)_{k=1}^K)$ is convex and l.s.c. On the other hand, for a fixed $\mathbb{Q}$, the functional $\widetilde{\mathcal{L}}(\mathbb{Q}, \cdot)$ is concave by (iii) in Proposition A.1. These observations allow us to apply Sion's minimax theorem [96, Theorem 3.4] to swap $\min$ and $\inf$ in (15) and obtain using (16)

$$
\begin{aligned}
\mathcal{L}^* &= \sup_{f_1, \ldots, f_K \in \mathcal{C}(\mathcal{Y})} \min_{\mathbb{Q} \in \mathcal{P}(\mathcal{Y})} \sum_{k=1}^K \lambda_k \left\{ \int_{\mathcal{X}_k} f_k^{C_k}(x_k) \mathrm{d}\mathbb{P}_k(x_k) + \int_{\mathcal{X}} f_k(y) \mathrm{d}\mathbb{Q}(y) \right\} \\
&= \sup_{f_1, \ldots, f_K \in \mathcal{C}(\mathcal{Y})} \left\{ \sum_{k=1}^K \lambda_k \int_{\mathcal{X}_k} f_k^{C_k}(x_k) \mathrm{d}\mathbb{P}_k(x_k) + \min_{\mathbb{Q} \in \mathcal{P}(\mathcal{Y})} \int_{\mathcal{X}} \bar{f}(y) \mathrm{d}\mathbb{Q}(y) \right\} \\
&= \sup_{f_1, \ldots, f_K \in \mathcal{C}(\mathcal{Y})} \underbrace{\left\{ \sum_{k=1}^K \lambda_k \int_{\mathcal{X}_k} f_k^{C_k}(x_k) \mathrm{d}\mathbb{P}_k(x_k) + \min_{y \in \mathcal{Y}} \bar{f}(y) \right\}}_{\stackrel{\text{def}}{=} \widetilde{\mathcal{L}}(f_1, \ldots, f_K)}.
\end{aligned} \tag{17}
$$

Next, we show that the $\sup$ in (17) can be restricted to tuplets satisfying the congruence condition $\sum_{k=1}^K \lambda_k f_k = 0$. It remains to show that for every tuplet $(f_1, \ldots, f_K) \in \mathcal{C}(\mathcal{Y})^K$ there exists a *congruent* tuplet $(\tilde{f}_1, \ldots, \tilde{f}_K) \in \mathcal{C}(\mathcal{Y})^K$ such that $\widetilde{\mathcal{L}}(\tilde{f}_1, \ldots, \tilde{f}_K) \geq \widetilde{\mathcal{L}}(f_1, \ldots, f_K)$.

To this end, fix $(f_1, \ldots, f_K)$ and define the congruent tuplet

$$
(\tilde{f}_1, \ldots, \tilde{f}_K) \stackrel{\text{def}}{=} \left( f_1, \ldots, f_{K-1}, f_K - \frac{\bar{f}}{\lambda_K} \right). \tag{18}
$$

We find $\tilde{M} \stackrel{\text{def}}{=} \inf_{y \in \mathcal{Y}} \sum_{k=1}^K \lambda_k \tilde{f}_k = 0$ by the congruence and derive

$$
\begin{aligned}
\widetilde{\mathcal{L}}(\tilde{f}_1, \ldots, \tilde{f}_K) - \widetilde{\mathcal{L}}(f_1, \ldots, f_K) &= \lambda_K \int_{\mathcal{X}_K} \left[ \tilde{f}_K^{C_K}(x_K) - f_K^{C_K}(x_K) \right] \mathrm{d}\mathbb{P}_K(x_K) - M \\
&\geq \lambda_K \int_{\mathcal{X}_K} \left[ \left( f_K - \frac{M}{\lambda_K} \right)^{C_K}(x_K) - f_K^{C_K}(x_K) \right] \mathrm{d}\mathbb{P}(x_K) - M \\
&= \lambda_K \int_{\mathcal{X}_K} \frac{M}{\lambda_K} \mathrm{d}\mathbb{P}(x_K) - M = 0,
\end{aligned}
$$

where the first inequality follows from $\tilde{f}_K = f_K - \frac{\bar{f}}{\lambda_K} \leq f_K - \frac{M}{\lambda_K}$ combined with monotonicity of the $C$-transform, see (i) in Proposition A.1. The second to last equality follow from constant additivity, see (ii) in Proposition A.1.

In summary, we obtain

$$
\mathcal{L}^* = \sup_{\substack{f_1, \ldots, f_k \in \mathcal{C}(\mathcal{Y}) \\ \sum_{k=1}^K f_k = 0}} \widetilde{\mathcal{L}}(f_1, \ldots, f_K). \tag{19}
$$

Finally, observe that for congruent $(f_1, \ldots, f_K)$ we have $\widetilde{\mathcal{L}}(f_1, \ldots, f_K) = \mathcal{L}(f_1, \ldots, f_K)$. Hence, we can replace $\widetilde{\mathcal{L}}$ by $\mathcal{L}$ in (19), which yields (6). $\qquad \square$

### A.3 Proof of Theorem 4.2

*Proof.* Write $\mathbb{Q}^*$ for the barycenter and $\pi_k^*$ for the optimizer of $\mathrm{EOT}_{c_k, \epsilon}(\mathbb{P}_k, \mathbb{Q}^*)$. Consider congruent potentials $f_1, \ldots, f_K \in \mathcal{C}(\mathcal{Y})$ and define the probability distribution

$$
\mathrm{d}\pi^{f_k}(x_k, y) \stackrel{\text{def}}{=} \mathrm{d}\mu_{x_k}^{f_k}(y) \, \mathrm{d}\mathbb{P}_k(x_k),
$$

where

$$
\frac{\mathrm{d}\mu_{x_k}^{f_k}(y)}{\mathrm{d}y} \stackrel{\text{def}}{=} \frac{1}{Z_{c_k}(f_k, x_k)} \exp\left( \frac{f_k(y) - c_k(x_k, y)}{\epsilon} \right), \tag{20}
$$

$$Z_{c_k}(f_k, x_k) \stackrel{\text{def}}{=} \log\left(\int_{\mathcal{Y}} e^{\frac{f_k(y) - c_k(x_k, y)}{\epsilon}} \mathrm{d}y\right). \tag{21}$$

Then we have by [78, Thm. 2]:

$$\mathrm{EOT}_{c_k, \epsilon}(\mathbb{P}_k, \mathbb{Q}^*) - \left(\int_{\mathcal{X}_k} f^{C_k}(x_k)\mathrm{d}\mathbb{P}(x_k) + \int_{\mathcal{Y}} f(y)\mathrm{d}\mathbb{Q}^*(y)\right) = \epsilon \mathrm{KL}\left(\pi_k^*\|\pi^{f_k}\right). \tag{22}$$

Multiplying (22) by $\lambda_k$ and summing over $k$ yields

$$\epsilon \sum_{k=1}^{K} \lambda_k \mathrm{KL}\left(\pi_k^*\|\pi^{f_k}\right) = \sum_{k=1}^{K} \lambda_k \left\{\mathrm{EOT}_{c_k, \epsilon}(\mathbb{P}_k, \mathbb{Q}^*) - \int_{\mathcal{X}_k} f_k^{C_k}(x_k)\mathrm{d}\mathbb{P}_k(x_k)\right\} - \int_{\mathcal{Y}} \underbrace{\sum_{k=1}^{K} \lambda_k f(y)}_{=0} \mathrm{d}\mathbb{Q}^*(y)$$

$$= \mathcal{L}^* - \mathcal{L}(f_1, \ldots, f_k), \tag{23}$$

where the last equality follows by congruence, i.e., $\sum_{k=1}^{K} \lambda_k f_k \equiv 0$.

The remaining inequality in (7) is a consequence of the data processing inequality for $f$-divergences which we invoke here to get

$$\mathrm{KL}\left(\pi_k^*\|\pi^{f_k}\right) \geq \mathrm{KL}\left(\mathbb{Q}^*\|\mathbb{Q}^{f_k}\right),$$

where $\mathbb{Q}^*$ and $\mathbb{Q}^{f_k}$ are the second marginals of $\pi_k^*$ and $\pi^{f_k}$, respectively. $\qquad\square$

### A.4 Proof of Theorem 4.3

*Proof.* The desired equation (9) could be derived exactly the same way as in [78, Theorem 3]. $\quad\square$

### A.5 Proof of Proposition 4.4 and Theorem 4.5

The derivations of the quantitative bound for Proposition 4.4 and Theorem 4.5 relies on the following standard definitions from learning theory, which we now recall for convenience (see, e.g. [93, §26]). Consider some class $\mathcal{S}$ of functions $s : \mathcal{X} \to \mathbb{R}$ and a distribution $\mu$ on $\mathcal{X}$. Let $X = \{x^1, \ldots, x^N\}$ be a sample of $N$ points in $\mathcal{X}$.

The **representativeness** of the sample $X$ w.r.t. the class $\mathcal{S}$ and the distribution $\mu$ is defined by

$$\mathrm{Rep}_X(\mathcal{S}, \mu) \stackrel{\text{def}}{=} \sup_{s \in \mathcal{S}} \left[\int_{\mathcal{X}} s(x)\mathrm{d}\mu(x) - \frac{1}{N}\sum_{n=1}^{N} s(x^n)\right]. \tag{24}$$

The **Rademacher complexity** of the class $\mathcal{S}$ w.r.t. the distribution $\mu$ and sample size $N$ is given by

$$\mathcal{R}_N(\mathcal{S}, \mu) \stackrel{\text{def}}{=} \frac{1}{N}\mathbb{E}\left\{\sup_{s \in \mathcal{S}} \sum_{n=1}^{N} s(x^n)\sigma_n\right\}, \tag{25}$$

where $\{x^n\}_{n=1}^{N} \sim \mu$ are mutually independent, $\{\sigma^n\}_{n=1}^{N}$ are mutually independent Rademacher random variables, i.e., $\mathrm{Prob}(\sigma^n = 1) = \mathrm{Prob}(\sigma^n = -1) = 0.5$, and the expectation is taken with respect to both $\{x_n\}_{n=1}^{N}, \{\sigma_n\}_{n=1}^{N}$. The well-celebrated relation between (25) and (24), as shown in [93, Lemma 26.2], is

$$\mathbb{E}\mathrm{Rep}_X(\mathcal{S}, \mu) \leq 2 \cdot \mathcal{R}_N(\mathcal{S}, \mu), \tag{26}$$

where the expectation is taken w.r.t. random i.i.d. sample $X \sim \mu$ of size $N$.

*Proposition 4.4.* Observe that by (23) we may write

$$\epsilon \sum_{k=1}^{K} \lambda_k \mathrm{KL}\left(\pi_k^*\|\pi^{\widehat{f}_k}\right) = \mathcal{L}^* - \mathcal{L}(\widehat{\mathbf{f}}) = \underbrace{\left[\mathcal{L}^* - \max_{\mathbf{f} \in \overline{\mathcal{F}}} \mathcal{L}(\mathbf{f})\right]}_{\text{Approximation error}} + \underbrace{\left[\max_{\mathbf{f} \in \overline{\mathcal{F}}} \mathcal{L}(\mathbf{f}) - \mathcal{L}(\widehat{\mathbf{f}})\right]}_{\text{Estimation error}}. \tag{27}$$

Let $\bar{\mathbf{f}}$ be a maximizer of $\mathcal{L}(\mathbf{f})$ within $\overline{\mathcal{F}}$. Analysing the estimation error in (27) yields

$$\max_{\mathbf{f} \in \overline{\mathcal{F}}} \mathcal{L}(\mathbf{f}) - \mathcal{L}(\widehat{\mathbf{f}}) = \mathcal{L}(\bar{\mathbf{f}}) - \mathcal{L}(\widehat{\mathbf{f}})$$

$$= \left[\mathcal{L}(\bar{\mathbf{f}}) - \widehat{\mathcal{L}}(\bar{\mathbf{f}})\right] + \underbrace{\left[\widehat{\mathcal{L}}(\bar{\mathbf{f}}) - \widehat{\mathcal{L}}(\widehat{\mathbf{f}})\right]}_{\leq 0} + \left[\widehat{\mathcal{L}}(\widehat{\mathbf{f}}) - \mathcal{L}(\widehat{\mathbf{f}})\right] \tag{28}$$

$$\leq 2 \sup_{\mathbf{f} \in \overline{\mathcal{F}}} \left|\mathcal{L}(\mathbf{f}) - \widehat{\mathcal{L}}(\mathbf{f})\right|, \tag{29}$$

where central term in line (28) is bounded above by 0 due the maximality of $\widehat{\mathbf{f}}$, that is, $\widehat{\mathcal{L}}(\widehat{\mathbf{f}}) = \max_{\mathbf{f} \in \overline{\mathcal{F}}} \widehat{\mathcal{L}}(\mathbf{f}) \geq \widehat{\mathcal{L}}(\bar{\mathbf{f}})$. Due to (29), we can bound the estimation error using the Rademacher complexity

$$\sup_{\mathbf{f} \in \overline{\mathcal{F}}} \left|\mathcal{L}(\mathbf{f}) - \widehat{\mathcal{L}}(\mathbf{f})\right| \leq \sum_{k=1}^{K} \lambda_k \sup_{f_k \in \mathcal{F}_k} \left[\int_{\mathcal{X}_k} f_k^{C_k}(x_k) \mathrm{d}\mathbb{P}_k(x_k) - \frac{1}{N_k} \sum_{n=1}^{N_k} f_k^{C_k}(x_k^n)\right] = \sum_{k=1}^{K} \lambda_k \mathrm{Rep}_{X_k}(\mathcal{F}_k^{C_k}, \mathbb{P}_k).$$

$\square$

*Proof of Theorem 4.5.* **Case (a) - Lipschitz costs**: Assume that, for $k \in \overline{K}$, $x \mapsto c_k(x, y)$ is Lipschitz with constant $L_k \geq 0$ for every $y \in \mathcal{Y}$. Recall that $f_k^{C_k}$ is defined as the pointwise supremum of $L_k$-Lipschitz functions and, therefore, Lipschitz continuous with the same constant. Since the value of the representativeness of a sample w.r.t. a function class is invariant under translating individual elements of said class, we have that $\mathrm{Rep}_X(\mathcal{F}_k^{C_k}, \mathbb{P}_k)$ coincides with $\mathrm{Rep}_X(\mathcal{G}_k, \mathbb{P}_k)$ where

$$\mathcal{G}_k \overset{\text{def}}{=} \{f^{C_k} - f^{C_k}(\tilde{x}_k) : f \in C(\mathcal{Y})\},$$

for some fixed $\tilde{x}_k \in \mathcal{X}_k$. All the functions in this class are $L_k$-Lipschitz and, therefore, bounded by $L_k \cdot \mathrm{diam}(\mathcal{X}_k)$. We may therefore apply [38, Theorem 4.3] to the class $\mathcal{G}_k$ and obtain

$$\mathbb{E}_{X \sim \mathbb{P}_k} \mathrm{Rep}_X(\mathcal{F}_k^{C_k}, \mathbb{P}_k) = \mathbb{E}_{X \sim \mathbb{P}_k} \mathrm{Rep}_X(\mathcal{G}_k, \mathbb{P}_k) \leq 2\mathcal{R}_N(\mathcal{G}_k, \mathbb{P}_k) \leq O(N_k^{-\frac{1}{D_k+1}}).$$

**Case (b) - Feature-based quadratic costs** $\frac{1}{2}\|u_k(\cdot) - v(\cdot)\|_2^2$: Alternatively, consider the case where $c_k(x_k, y) = \frac{1}{2}\|u_k(x_k) - v(y)\|_2^2$ and $\mathcal{F}_k \subseteq C(\mathcal{Y})$ is bounded (w.r.t. the supremum norm).

To this end, recall that for a measurable and bounded function $f : \mathcal{Y} \to \mathbb{R}$, the weak entropic $c_k$-transform satisfies

$$f^{C_k}(x_k) = -\epsilon \log(Z_{c_k}(f, x_k)),$$

where $Z_{c_k}(f, x_k) = \int_{\mathcal{Y}} \exp\left(\frac{f(y) - c_k(x_k, y)}{\epsilon}\right) dy$. Recall that $\mathbb{R}^{D''} \times \mathbb{R}^{D''} \ni (a, b) \mapsto \exp\left(-\frac{\|a-b\|^2}{2\epsilon}\right)$ is a positive definite kernel which is widely known as the **Gaussian kernel**. This means that there exists a Hilbert space $\mathcal{H}$ and a feature map $\phi : \mathbb{R}^{D''} \to \mathcal{H}$ such that $\exp\left(-\frac{\|a-b\|^2}{2\epsilon}\right) = \langle\phi(a), \phi(b)\rangle_{\mathcal{H}}$. Due to the particular form of $c_k$, we may write

$$\exp\left(-\frac{c_k(x_k, y)}{\epsilon}\right) = \exp\left(-\frac{\|u_k(x) - v(y)\|^2}{2\epsilon}\right) = \langle\phi(u_k(x_k)), \phi(v(y))\rangle_{\mathcal{H}}. \tag{30}$$

Notice that $\{\phi(u_k(x_k)) : x_k \in \mathcal{X}_k\} \subseteq \{v \in \mathcal{H} : \|v\|_{\mathcal{H}} = 1\}$ since $\|\phi(a)\|_{\mathcal{H}}^2 = \langle\phi(a), \phi(a)\rangle_{\mathcal{H}} = 1$ for every $a \in \mathbb{R}^{D''}$.

Using the identity in (30), we can express $Z_{c_k}(f, x_k)$ by

$$Z_{c_k}(f, x_k) = \int_{\mathcal{Y}} \langle\phi(u_k(x_k)), \phi(v(y))\rangle_{\mathcal{H}} \, e^{\frac{f(y)}{\epsilon}} \, dy = \left\langle\phi\big(u_k(x_k)\big), \int_{\mathcal{Y}} \phi(v(y)) e^{\frac{f(y)}{\epsilon}} \, dy\right\rangle_{\mathcal{H}},$$

where the last equality is justified as $\mathcal{Y}$ is compact; furthermore, we note the integrals are well-defined by the measurability of $\phi$, $u_k$ and $v$. Moreover, using the boundedness of each $\mathcal{F}_k$ and compactness of $\mathcal{Y}$, we get

$$R \overset{\text{def}}{=} \max_{k=1,\dots,K} \sup_{f \in \mathcal{F}_k} \left\|\int_{\mathcal{Y}} \phi(v(y)) e^{\frac{f(y)}{\epsilon}} \, dy\right\|_{\mathcal{H}} < \infty. \tag{31}$$

Define $\mathcal{G}_k \overset{\text{def}}{=} \{Z_{c_k}(f, \cdot) : f \in \mathcal{F}_k\}$ and observe that $\mathcal{G}_k \subseteq \mathcal{G}_k' \overset{\text{def}}{=} \{\langle\phi(u_k(\cdot)), w\rangle_{\mathcal{H}} : \|w\|_{\mathcal{H}} \leq R\}$. This implies that $\mathcal{R}_{N_k}(\mathcal{G}_k, \mathbb{P}_k) \leq \mathcal{R}_{N_k}(\mathcal{G}_k', \mathbb{P}_k)$ by the properties of the Rademacher complexity.

In turn, the latter Rademacher complexity can be bounded by [10, Lemma 22]. Indeed, write $\mathbb{P}'_k \stackrel{\text{def}}{=} (u_k)_\# \mathbb{P}_k$ such that

$$\mathcal{R}_{N_k}(\mathcal{G}'_k, \mathbb{P}_k) = \mathcal{R}_{N_k}(\{\langle \phi(\cdot), w \rangle_\mathcal{H} : \|w\|_\mathcal{H} \le R\}, \mathbb{P}'_k),$$

which can be directly seen from the definition of the Rademacher complexity. Thus, we can apply [10, Lemma 22] to the right-hand side, and summarizing obtain

$$\mathcal{R}_{N_k}(\mathcal{G}_k, \mathbb{P}_k) \le \frac{R}{\sqrt{N_k}}.$$

Since the functions in $\mathcal{G}_k$ are bounded uniformly away from zero by some constant $\kappa > 0$ (depending on the bound of $\mathcal{F}_k$ and $\epsilon$), and since the logarithm restricted to $[\kappa, \infty)$ is $1/\kappa$-Lipschitz, we have that

$$\mathcal{R}_{N_k}(\mathcal{F}_k^{C_k}, \mathbb{P}_k) \le \frac{\epsilon}{\kappa} \mathcal{R}_{N_k}(\mathcal{G}_k, \mathbb{P}_k) \le \frac{\epsilon}{\kappa} \frac{R}{\sqrt{N_k}}.$$

We can now bound the expected representativeness of $\mathcal{F}_k^{C_k}$ with the Rademacher complexity by (26), yielding the claim. $\qquad \square$

## A.6 Proof of Theorem 4.6

*Proof of Theorem 4.6.* Let $\sigma \in C(\mathbb{R})$ be a non-affine activation function which is differentiable at some point $x_0 \in \mathbb{R}$ and for which $\sigma'(x_0) \ne 0$. Let $\delta > 0$, $\lambda_1, \dots, \lambda_K > 0$, and $K \in \mathbb{N}$ be given. By Theorem 4.1, there exist $K$ congruent continuous functions $f'_1, \dots, f'_K$ such that

$$\mathcal{L}^* - \mathcal{L}(\tilde{f}_1, \dots, \tilde{f}_K) < \frac{\delta}{2}. \tag{32}$$

Applying [29, Theorem 4.1] we deduce that for each $k = 1, \dots, K$, there exist a continuous extension $f'_k : \mathbb{R}^D \to \mathbb{R}$ for $\tilde{f}_k$ to all of $\mathbb{R}^D$; i.e. $f'_k(y) = \tilde{f}_k(y)$ for each $y \in \mathcal{Y}$. In particular, (32) can be rewritten as

$$\mathcal{L}^* - \mathcal{L}(f'_1, \dots, f'_K) < \frac{\delta}{2}. \tag{33}$$

Set $\delta_k \stackrel{\text{def}}{=} \delta/(4\lambda_k)$ for each $k = 1, \dots, K$. Since $\mathcal{Y}$ is a compact subset of $\mathbb{R}^D$, and since we have assumed that $\sigma \in C(\mathbb{R})$ is non-affine activation function which is differentiable at some point $\tilde{x} \in \mathbb{R}$ and for which $\sigma'(\tilde{x}) \ne 0$ then the special case of [62, Theorem 9] given in [62, Proposition 53], implies that for any there exist feedforward neural networks $g_k : \mathbb{R}^{D_k} \to \mathbb{R}$ ($k \in \overline{K}$) with activation function $\sigma$, such that

$$\|g_k - f'_k\|_\infty \stackrel{\text{def}}{=} \sup_{y \in \mathcal{Y}} |g_k(y) - f'_k(y)| = \sup_{y \in \mathcal{Y}} |g_k(y) - f'_k(y)| < \delta_k,$$

each $g_k$ width at-most $D + 4$. Pick $\delta_k = \frac{\delta}{4}$ for all $k \in \overline{K}$ and suitable neural networks $g_1, \dots, g_K$. Next, we define the congruent sums of neural networks $f_k \stackrel{\text{def}}{=} g_k - \sum_{k=1}^K \lambda_k g_k$. We derive

$$\Big\| \sum_{k=1}^K \lambda_k g_k \Big\|_\infty = \Big\| \sum_{k=1}^K \lambda_k g_k - \underbrace{\sum_{k=1}^K \lambda_k f'_k}_{=0} \Big\|_\infty \le \sum_{k=1}^K \lambda_k \big\| g_k - f'_k \big\|_\infty < \sum_{k=1}^K \lambda_k \frac{\delta}{4\lambda_k} = \frac{\delta}{4}. \tag{34}$$

Using (34) we obtain for fixed $k \in \overline{K}$

$$\|f'_k - f_k\|_\infty = \|f'_k - g_k + \sum_{k'=1}^K \lambda_{k'} g_{k'}\|_\infty \le \underbrace{\|f'_k - g_k\|_\infty}_{<\frac{\delta}{4}} + \underbrace{\|\sum_{k'=1}^K \lambda_{k'} g_{k'}\|_\infty}_{<\frac{\delta}{4}} < \frac{\delta}{2}. \tag{35}$$

By (iv) in Proposition A.1 together with (35) we find

$$\|f_k^{C_k} - (f'_k)^{C_k}\|_\infty \le \|f_k - f'_k\|_\infty < \frac{\delta}{2}. \tag{36}$$

Now we use (36) to derive

$$|\mathcal{L}(f_1,\ldots,f_K) - \mathcal{L}(f_1',\ldots,f_K')| \leq \sum_{k=1}^K \lambda_k \left| \int_{\mathcal{X}_k} f_k^{C_k}(x_k)\mathrm{d}\mathbb{P}_k(x_k) - \int_{\mathcal{X}_k} (f_k')^{C_k}(x_k)\mathrm{d}\mathbb{P}_k(x_k) \right|$$

$$\leq \sum_{k=1}^K \lambda_k \int_{\mathcal{X}_k} |f_k^{C_k}(x_k) - (f_k')^{C_k}(x_k)|\mathrm{d}\mathbb{P}_k(x_k)$$

$$\leq \sum_{k=1}^K \lambda_k \|f_k^{C_k} - (f_k')^{C_k}\|_\infty < \underbrace{\left(\sum_{k=1}^K \lambda_k\right)}_{=1} \frac{\delta}{2} = \frac{\delta}{2}. \tag{37}$$

Next we combine (33) with (37) to get

$$\mathcal{L}^* - \mathcal{L}(f_1,\ldots,f_K) \leq \underbrace{[\mathcal{L}^* - \mathcal{L}(f_1',\ldots,f_K')]}_{<\delta/2} + \underbrace{|\mathcal{L}(f_1,\ldots,f_K) - \mathcal{L}(f_1',\ldots,f_K')|}_{<\delta/2} < \delta. \tag{38}$$

By using (38) together with Theorem 4.2 we obtain

$$\epsilon \sum_{k=1}^K \lambda_k \mathrm{KL}\left(\pi_k^* \| \pi^{f_k}\right) = \mathcal{L}^* - \mathcal{L}(f_1,\ldots,f_K) < \delta$$

which completes the proof. $\qquad\square$

### A.7 Existence and uniqueness of the barycenter distribution which solves (5)

We introduce an auxiliary functional which is the argument of minimization problem (5):

$$\mathcal{B}(\mathbb{Q}) \stackrel{\text{def}}{=} \sum_{k=1}^K \lambda_k \mathrm{EOT}_{c_k,\epsilon}(\mathbb{P}_k, \mathbb{Q}), \tag{39}$$

i.e. the optimal value of (5) could be defined as $\mathcal{L}^* = \inf\limits_{\mathbb{Q} \in \mathcal{P}(\mathcal{Y})} \mathcal{B}(\mathbb{Q})$.

Note that the functional $\mathbb{Q} \mapsto \mathcal{B}(\mathbb{Q})$ is strictly convex and lower semicontinuous (w.r.t. the weak topology) as each component $\mathbb{Q} \mapsto \mathrm{EOT}_{c_k,\epsilon}(\mathbb{P}_k, \mathbb{Q})$ is strictly convex and l.s.c. (lower semi-continuous) itself. The latter follows from [9, Th. 2.9] by noting that on $\mathcal{P}(\mathcal{Y})$ the map $\mu \mapsto \mathbb{E}_{y\sim\mu} c_k(x,y) - H(\mu)$ is l.s.c, bounded from below and strictly convex thanks to the entropy term. Since $\mathcal{P}(\mathcal{Y})$ is weakly compact (as $\mathcal{Y}$ is compact due to Prokhorov's theorem, see, e.g., [89, Box 1.4]), it holds that $\mathcal{B}(\mathbb{Q})$ admits at least one minimizer due to the Weierstrass theorem [89, Box 1.1], i.e., a barycenter $\mathbb{Q}^*$ exists. In the paper, *we work under the reasonable assumption that there exists at least one $\mathbb{Q}$ for which $\mathcal{B}(\mathbb{Q}) < \infty$.* In this case, the barycenter $\mathbb{Q}^*$ is unique due to the strict convexity of $\mathcal{B}$.

## B Extended discussions

### B.1 Extended discussion of related works

**Discrete OT-based solvers** provide solutions to OT-related problems between discrete distributions. A comprehensive overview can be found in [83]. The discrete OT methods for EOT barycenter estimation are [20, 98, 11, 21, 22, 30, 63, 49, 68]. An alternative formulation of the barycenter problem based on weak mass transport and corresponding discrete solver could be found in [12]. In spite of sound theoretical foundations and established convergence guarantees [64], these approaches can not be directly adapted to our learning setup, see §2.3.

**Continuous OT solvers.** Beside the continuous EOT solvers discussed in §3, there exist a variety of neural OT solver for the non-entropic (unregularized, $\epsilon = 0$) case. For example, solvers such as [44, 74, 54, 58, 57, 32, 35, 88, 5, 31], are based on optimizing the dual form, similar to our (2), with neural networks. We mention these methods because they serve as the basis for certain continuous unregularized barycenter solvers. For example, ideas of [55] are employed in the barycenter method presented in [59]; the solver from [74] is applied in [32]; max-min solver introduced in [58] is used

in [55]. It is also worth noting that there exist several neural solvers that cater to more general OT problem formulations [61, 60, 31, 8, 84]. These can even be adapted to the EOT case [42] but require substantial technical effort and the usage of restrictive neural architectures.

**Continuous EOT solvers** aim to recover the optimal EOT plan $\pi^*$ in EOT problems like (1) between unknown distributions $\mathbb{P}$ and $\mathbb{Q}$ which are only accessible through a limited number of samples. One group of methods [37, 91, 23] is based on the dual formulation of OT problem regularized with KL divergences [37, Eq. $(\mathcal{P}_\epsilon)$] which is equivalent to (1). Another group of methods [109, 24, 13, 41, 106, 94] takes advantage of the dynamic reformulation of EOT via Schrödinger bridges [70, 76]. In turn, [51] solve EOT with conditional flow matching [106].

In [78], the authors propose an approach to tackle (1) by means of Energy-Based models [69, 100, EBM]. They develop an optimization procedure resembling standard EBM training which retrieves the optimal dual potential $f^*$ appearing in (2). As a byproduct, they recover the optimal conditional plans $\mu_x^{f^*} = \pi^*(\cdot|x)$. Our approach for solving the EOT barycenter (5) is primarily inspired by this work. In fact, we manage to overcome the theoretical and practical difficulties that arise when moving from the EOT problem guided with EBMs to the EOT barycenter problem (multiple marginals, optimization with respect to an *unfixed* marginal distribution $\mathbb{Q}$, see §4 for details of our method.

**Other related works.** Another relevant work is [95], where the authors study the barycenter problem and restrict the search space to a manifold produced by a GAN. This idea is also utilized in §5.2 and §5.3, but their overall setting drastically differs from our setup and actually is not applicable. We search for a barycenter of $K$ *high-dimensional image distributions* represented by their random samples (datasets). In contrast, they consider $K$ images, represent each *image as a $2D$ distribution* via its intensity histogram and search for a **single image** on the GAN manifold whose density is the barycenter of the input images. To compute the barycenter, they use discrete OT solver. In summary, neither our barycenter solver is intended to be used in their setup, nor their method is targeted to solve the problems considered in our paper.

## B.2 Extended discussion of potential applications

It is not a secret that despite considerable efforts in developing continuous barycenter solvers [72, 59, 55, 32, 82, 14], these solvers have not found yet a real working practical application. The reasons for this are two-fold:

1. Existing continuous barycenter solvers (Table 1) are yet not scalable enough and/or work exclusively with the quadratic cost ($\ell^2$), which might be not sufficient for the practical needs.

2. Potential applications of barycenter solvers are too technical and, unfortunately, require substantial efforts (challenging and costly data collection, non-trivial design of task-specific cost functions, unclear performance metrics, etc.) to be implemented in practice.

Despite these challenges, there exist rather inspiring practical problem formulations where the continuous barycenter solvers may potentially shine and we name a few below. These potential applications motivate the research in the area. More generally, we hope that our developed solver could be a step towards applying continuous barycenters to practical tasks that benefit humanity.

**1. Solving domain shift problems in medicine (Fig. 6a).** In medicine, it is common that the data is collected from multiple sources (laboratories, clinics) and using different equipment from various vendors, each with varying technical characteristics [40, 65, 53, 103, 112]. Moreover, the data coming from each source may be of limited size. These issues complicate building robust and reliable machine learning models by using such datasets, e.g., learning MRI segmentation models to assist doctors.

A potential way to overcome the above-mentioned limitations is to find a common representation of the data coming from multiple sources. This representation would require translation maps that can transform the new (previously unseen) data from each of the sources to this shared representation. This formulation closely aligns with the continuous barycenter learning setup (§2.3) studied in our paper. In this context, the barycenter could play the role of the shared representation.

To apply barycenters effectively to such domain averaging problems, two crucial ingredients are likely required: appropriate cost functions $c_k$ and a suitable data manifold $\mathcal{M}$ in which to search for the barycenters. The design of the cost itself may be a challenge requiring certain domain-specific knowledge that necessitates involving experts in the field. Meanwhile, the manifold constraint is

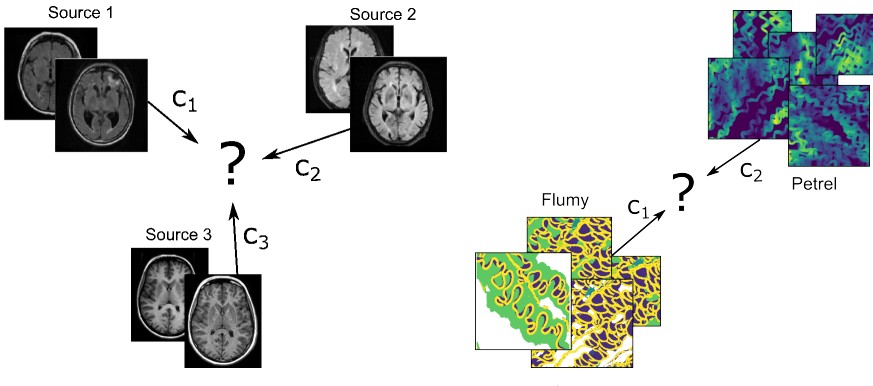

(a) Domain averaging of MRI scans' sources.    (b) Mixing geological simulators.

Figure 6: A schematic presentation of potential applications of barycenter solvers.

required to avoid practically meaningless barycenters such as those considered in §5.2. Nowadays, with the rapid growth of the field of generative models, it is reasonable to expect that soon the new large models targeted for medical data may appear, analogously to DALL-E [85], StableDiffusion [87] or StyleGAN-T [90] for general image synthesis. These future models could potentially parameterize the medical data manifolds of interest, opening new possibilities for medical data analysis.

**2. Mixing geological simulators (Fig. 6b).** In geological modeling, variuos simulators exist to model different aspects of underground deposits. Sometimes one needs to build a generic tool which can take into account several desired geological factors which are successfully modeled by separate simulators.

**Flumy**[1] is a process-based simulator that uses hydraulic theory [48] to model specific channel depositional processes returning a detailed three-dimensional geomodel informed with deposit lithotype, age and grain size. However, its result is a 3D segmentation field of facies (rock types) and it does not produce the real valued porosity field needed for hydrodynamical modeling.

**Petrel**[2] software is the other popular simulator in the oil and gas industry. It is able to model complex real-valued geological maps such as the distribution of porosity. The produced porosity fields may not be realistic enough due to paying limited attention to the geological formation physics.

Both Flumy and Petrel simulators contain some level of stochasticity and are hard to use in conjunction. Even when conditioned on common prior information about the deposit, they may produce maps of facies and permeability which do not meaningfully correspond to each other. This limitation provides potential *prospects for barycenter solvers* which could be used to *get the best from both simulators* by mixing the distributions produced by each of them.

From our personal discussions with the experts in the field of geology, such task formulations are of considerable interest both for scientific community as well as industry. Applying our barycenter solver in this context is a challenge for future research. We acknowledge that this would also require overcoming considerable technical and domain-specific issues, including the data collection and the choice of costs $c_k$.

### B.3 Extended discussion on doubly-regularized OT barycenters

The objective (5) is not the only way to formulate Entropic OT barycenter problem. Recent studies [15, 82] consider the so-called *doubly*-regularized Entropic OT barycenters. Following the notations from [15], for $\lambda \geq 0$ and $\tau \geq 0$ we define:

$$\text{EOT}^{\text{dr}}_{c,\lambda,\tau}(\mathbb{P},\mathbb{Q}) \stackrel{\text{def}}{=} \min_{\pi \in \Pi(\mathbb{P},\mathbb{Q})} \left\{ \underset{(x,y)\sim\pi}{\mathbb{E}}\, c(x,y) + \lambda KL(\pi\|\mathbb{P}\otimes\mathbb{Q}) + \tau H(\mathbb{Q}) \right\}.$$

The corresponding *doubly*-regularized EOT barycenter problem is as follows:

$$\mathcal{L}^{*,\text{dr}} \stackrel{\text{def}}{=} \inf_{\mathbb{Q}\in\mathcal{P}(\mathcal{Y})} \mathcal{B}^{\text{dr}}(\mathbb{Q}) \stackrel{\text{def}}{=} \inf_{\mathbb{Q}\in\mathcal{P}(\mathcal{Y})} \sum_{k=1}^{K} \lambda_k \text{EOT}^{\text{dr}}_{c,\lambda,\tau}(\mathbb{P}_k,\mathbb{Q}). \tag{40}$$

---

[1] https://flumy.minesparis.psl.eu

[2] https://www.software.slb.com/products/petrel

It turns out that our considered EOT barycenter (5) with regularization strength $\epsilon$ is the particular case of (40) with $\lambda = \tau = \epsilon$. According to the classification [15, Table 1], this case is known as *Schrödinger* barycenter. The natural question arizes: Is it possible to adapt our energy-guided methodology for the case $\lambda \neq \tau$?

To answer this question, in what follows, we have a closer look at the differences between general *doubly*-regularized formulations and our particular *Schrödinger* specification. The important property of our case is that the entropy term $H(\mathbb{Q})$ completely disappears from the barycenter objective. In all the other regularized cases, when $\lambda \neq \tau$, this entropy term immediately reappears (either with the plus or minus sign). The presence of $H(\mathbb{Q})$ term notably differs from ours and seems like to be not suitable for our solver. In what follows, we explain the reasons.

In the *Schrödinger* case, the barycenter problem can be solved via optimizing conditional distributions $\pi_k(y|x_k)$. Namely, we use potentials $f_k$ combined with costs $c_k$ to approximate the energy functions (unnormalized log-densities) of these conditionals. We employ EBM-based techniques to compute the gradient of the learning objective which avoids the direct computation of the entropy terms $H(\pi_k(y|x_k))$ appearing in the $C_k$-transforms.

If we further add the non-zero term $H(\mathbb{Q})$ to the barycenter objective, this will presumably require (in the dual objective) a separate computation of the entropy terms $H(\pi_k(y))$ of second marginals $\pi_k(y)$ of each $\pi_k$. **This is highly non-trivial** and it seems like our *solver does not easily generalize to this case*. In our framework, we can get samples of $\pi_k(y)$ by MCMC (via sampling $x_k \sim \mathbb{P}_k$ and then running MCMC for $\pi(y|x_k)$). However, estimation of entropy of $\pi_k(y)$ from raw samples still remains infeasible. In particular, EBM-like techniques (which we employ) can not be used here to

| Experiment | training time | inference time |
|---|---|---|
| Toy $2D$ (**Ours**) | 3h | 20s |
| Gaussians (**Ours**) | 6h | 40s |
| Sphere (**Ours**) | 3h | 20s |
| MNIST 0/1 manifold (**Ours**) | 10h | 1m |
| MNIST 0/1 data (**Ours**) | 20h | 1m |
| Ave Celeba manifold (**Ours**) | 60h | 2m |
| Ave Celeba data (**Ours**) | 60h | 2m |
| Ave Celeba data (**WIN**) | 160h | 10s |
| Ave Celeba data (**SCWB**) | 200h | 10s |

Table 3: Computational complexity for **Ours** (all experiments) and baselines (Ave Celeba).

derive the gradient of the objective. This is because the required unnormalized density of $\pi_k(y)$ is itself unknown (we only know it for conditional distributions $\pi_k(y|x)$).

## C  Experimental Details

The hyperparameters of our solver are summarized in Table 4. Some hyperparameters, e.g., $L, S, iter$, are chosen primarily from time complexity reasons. Typically, the increase in these numbers positively affects the quality of the recovered solutions, see, e.g., [42, Appendix E, Table 16]. However, to reduce the computational burden, we report the figures which we found to be reasonable. Working with the manifold-constraint setup, we parameterize each $g_{\theta_k}(z)$ in our solver as $h_{\theta_k} \circ G(z)$, where $G$ is a pre-trained (frozen) StyleGAN and $h_{\theta_k}$ is a neural network with the ResNet architecture. We empirically found that this strategy provides better results than a direct MLP parameterization for the function $g_{\theta_k}(z)$.

| Experiment | $D$ | $K$ | $\epsilon$ | $\lambda_1$ | $\lambda_2$ | $\lambda_3$ | $\lambda_4$ | $f_{\theta,k}$ | $lr_{g_{\theta,k}}$ | $iter$ | $\sqrt{\eta}$ | $L$ | $S$ |
|---|---|---|---|---|---|---|---|---|---|---|---|---|---|
| Toy 2D | 2 | 3 | $10^{-2}$ | 1/3 | 1/3 | 1/3 | - | MLP | $10^{-2}$ | 200 | 1.0 | 300 | 256 |
| MNIST 0/1 | 1024 | 2 | $10^{-2}$ | 0.5 | 0.5 | - | - | ResNet | $10^{-4}$ | 1000 | 0.1 | 500 | 32 |
| MNIST 0/1 | 512 | 2 | $10^{-2}$ | 0.5 | 0.5 | - | - | ResNet | $10^{-4}$ | 1000 | 0.1 | 250 | 32 |
| Ave, celeba! (Data) | $3*64^2$ | 3 | $10^{-2}$ | 0.25 | 0.5 | 0.25 | - | ResNet | $10^{-4}$ | 5000 | 0.1 | 500 | 64 |
| Ave, celeba! (Manifold) | 512 | 3 | $10^{-4}$ | 0.25 | 0.5 | 0.25 | - | ResNet | $10^{-4}$ | 1000 | 0.1 | 250 | 128 |
| Gaussians | 2-64 | 3 | $10^{-2}, 1$ | 0.25 | 0.25 | 0.5 | - | MLP | $10^{-3}$ | 50000 | 0.1 | 700 | 1024 |
| Sphere | 3 | 4 | $10^{-2}$ | 0.25 | 0.25 | 0.25 | 0.25 | MLP | $3 \times 10^{-3}$ | 1000 | 1.0 | 300 | 256 |
| Singlle-cell | 1000 | 2 | $10^{-2}$ | 0.25 | 0.75 | - | - | MLP | $5 \times 10^{-4}$ | 1000 | 0.05 | 1000 | 1024 |

Table 4: Hyperparameters that we use in the experiments with our Algorithm 1.

To train the StyleGAN for MNSIT01 & Ave, celeba! experiments, we employ the official code from

**Computational complexity.** We report the (approximate) time for training and inference (batch size $S = 128$) of our method on different experimental setups, see Table 3 (the hardware is a single V100 gpu). For Ave Celeba experiment, we additionally report the computational complexity of the competitors. As we can see, all the methods in this experiment (Ave Celeba) require a comparable amount of time for training. The inference with our approach is expectedly costlier than competitors due to the reliance on MCMC.

## C.1 Barycenters of 2D/3D Distributions

**Cartesian representation of twister map.** In §5.1 we define twister map $u$ using polar coordinate system. For clearness, we give its form on cartesian coordinates. Let $x \in \mathbb{R}^2$, $x = (x_{(1)}, x_{(2)})$. Note that $\|x\| = \sqrt{x_{(1)}^2 + x_{(2)}^2}$ Then,

$$u(x) = u(x_{(1)}, x_{(2)}) = \begin{pmatrix} \cos(\|x\|) & -\sin(\|x\|) \\ \sin(\|x\|) & \cos(\|x\|) \end{pmatrix} \begin{pmatrix} x_{(1)} \\ x_{(2)} \end{pmatrix},$$

i.e., the twister map rotates input points $x$ by angles equal to $\|x\|$.

**Analytical barycenter distribution for 2D Twister experiment.** Below, we provide a mathematical derivation that the true unregularized barycenter of the distributions $\mathbb{P}_1, \mathbb{P}_2, \mathbb{P}_3$ in Fig. 2a coincides with a Gaussian.

We begin with a rather general observation. Consider $\mathcal{X}_k = \mathcal{Y} = \mathbb{R}^D$ ($k \in \overline{K}$) and let $\mathrm{OT}_c \stackrel{\text{def}}{=} \mathrm{EOT}_{c,0}$ denote the unregularized OT problem ($\epsilon = 0$) for a given continuous cost function $c$. Let $u : \mathbb{R}^D \to \mathbb{R}^D$ be a measurable bijection and consider $\mathbb{P}'_k \in \mathcal{P}(\mathbb{R}^D)$ for $k \in \overline{K}$. By using the change of variables formula, we have for all $\mathbb{Q}' \in \mathcal{P}(\mathbb{R}^D)$ that

$$\mathrm{OT}_{c \circ (u \times u)}(u_{\#}^{-1}(\mathbb{P}'), u_{\#}^{-1}(\mathbb{Q}')) = \mathrm{OT}_c(\mathbb{P}', \mathbb{Q}'), \tag{41}$$

where $\#$ denotes the pushforward operator of distributions and $[c \circ (u \times u)](x, y) = c(u(x), u(y))$. Note that by (41) the barycenter of $\mathbb{P}'_1, \ldots, \mathbb{P}'_K$ for the unregularized problem with cost $c$ coincides with the result of applying the pushforward operator $u_{\#}^{-1}$ to the barycenter of the very same problem but with cost $c \circ (u \times u)$.

Next, we fix $u$ to be the twister map (§5.1). In Fig. 2a we plot the distributions $\mathbb{P}_1 \stackrel{\text{def}}{=} u_{\#}^{-1}\mathbb{P}'_1, u_{\#}^{-1}\mathbb{P}'_1, u_{\#}^{-1}\mathbb{P}'_3$ which are obtained from Gaussian distributions $\mathbb{P}'_1 = \mathcal{N}((0, 4), I_2), \mathbb{P}'_2 = \mathcal{N}((-2, 2\sqrt{3}), I_2), \mathbb{P}'_3 = \mathcal{N}((2, 2\sqrt{3}), I_2)$ by the pushforward. Here $I_2$ is the 2-dimensional identity matrix. For the unregularized $\ell^2$ barycenter problem, the barycenter of such shifted Gaussians can be derived analytically [3]. The solution coincides with a zero-centered standard Gaussian $\mathbb{Q}' = \mathcal{N}(0, I_2)$. Hence, the barycenter of $\mathbb{P}_1, \ldots, \mathbb{P}_K$ w.r.t. the cost $\ell^2 \circ (u \times u)$ is given by $\mathbb{Q}^* = u_{\#}^{-1}\mathbb{Q}'$. From the particular choice of $u$ it is not hard to see that $\mathbb{Q}^* = \mathbb{Q}' = \mathcal{N}(0, I_2)$ as well.

## C.2 Barycenters of MNIST Classes

**Additional qualitative examples** of our solver's results are given in Figure 7.

**Details of the baseline solvers.** For the solver by [32, SCWB], we run their publicly available code from the official repository

The authors do no provide checkpoints, and we train their barycenter model from scratch. In turn, for the solver by [55, WIN], we also employ the publicly available code

Here we do not train their models but just use the checkpoint available in their repo.

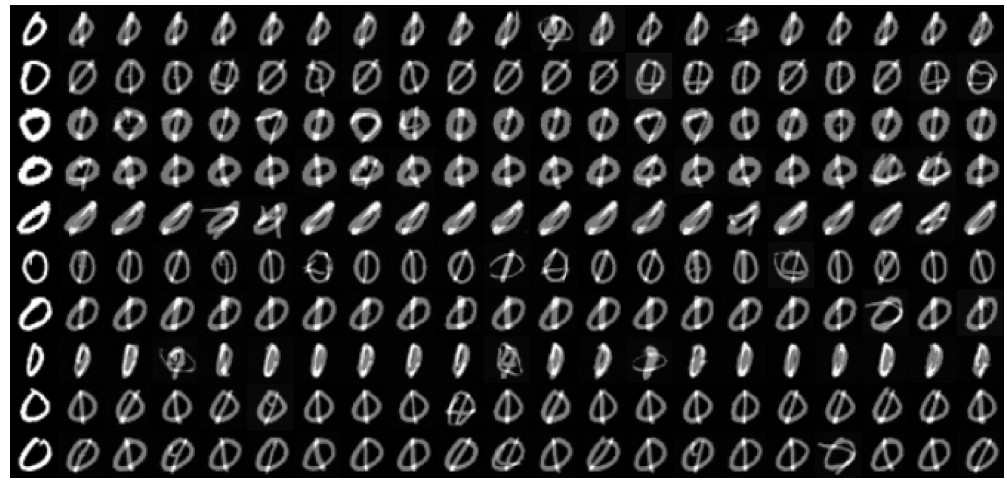

(a) Learned plans from $\mathbb{P}_1$ (zeros, 1st column) to the barycenter.

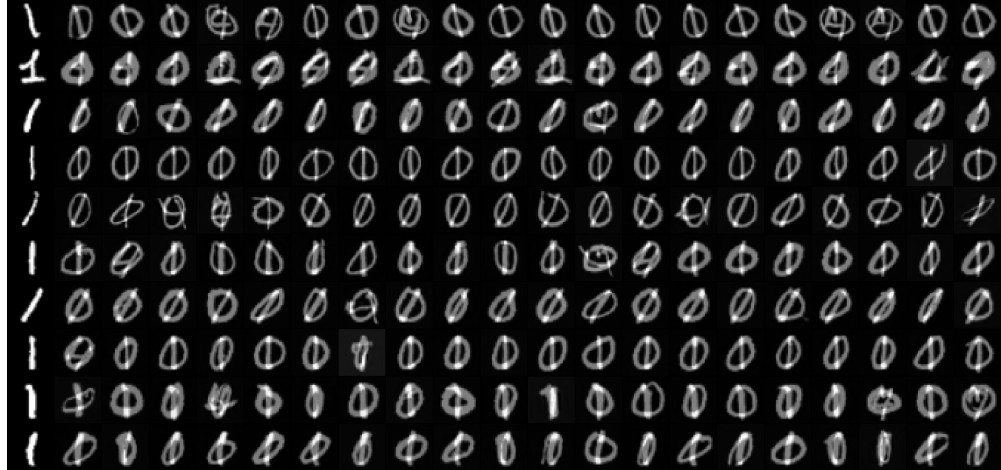

(b) Learned plans from $\mathbb{P}_2$ (ones, first column) to the barycenter.

Figure 7: *Experiment with averaging MNIST 0/1 digit classes.* The plot shows additional examples of samples transported with **our** solver to the barycenter.

### C.3   Barycenters of the Ave, Celeba! Dataset

**Additional qualitative examples** of our solver's results are given in Figure 8.

**Extra experiment in Data Space.** Our method in the manifold constrained setup on MNIST dataset performs better than in the data space setup (see Figure 5). It generates less noised images and demonstrates better perceptual quality. For this reason, we provide only Manifold-constrained setting for experiment with Ave, Celeba! dataset in §5.3.

For completeness, here we also test our method directly in the data space setup for Ave, Celeba! dataset. Hyperparameters of our solver are listed in Table 4. In Figure 10, we show images obtained in Data space setup. As expected, the FID scores in Data space are not that good as the images are more noised since we solve the entropy-regularized problem. But we stress one more time that our method permits StyleGAN manifold trick, which greatly improves the performance, see the images in Figure 4 for the manifold-constrained setup.

**Convergence at training.** We provide the behaviour of $\mathcal{L}_2$-UVP metric (between the unregularized ground truth barycenter mapping and the entropic barycenter mapping for our learned $\pi^{\tilde{f}_k}(y|x_k)$) for Ave Celeba experiment §5.3, see Figure 9 . We emphasize that $\mathcal{L}_2$-UVP directly measures (by computing pointwise MSE) how the learned mapping differs from the true mapping.

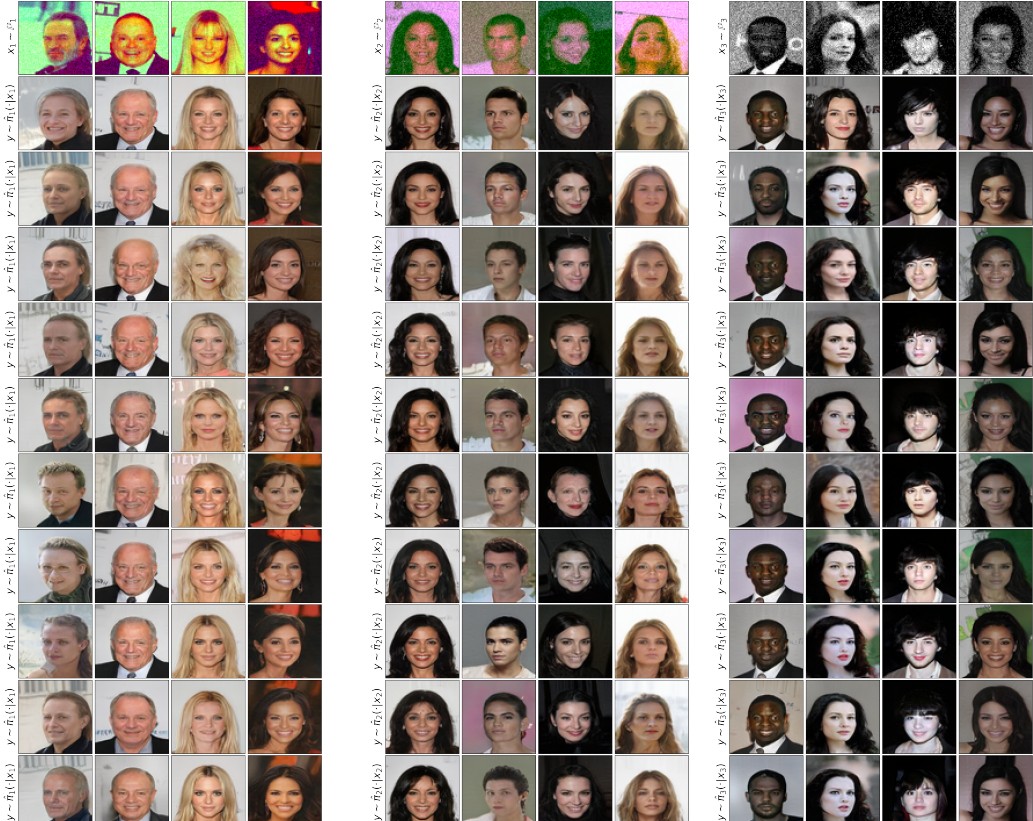

(a) Maps from $\mathbb{P}_1$ to the barycenter. (b) Maps from $\mathbb{P}_2$ to the barycenter. (c) Maps from $\mathbb{P}_3$ to the barycenter.

Figure 8: *Experiment on the Ave, celeba! barycenter dataset.* The plots show additional examples of samples transported with **our** solver to the barycenter.

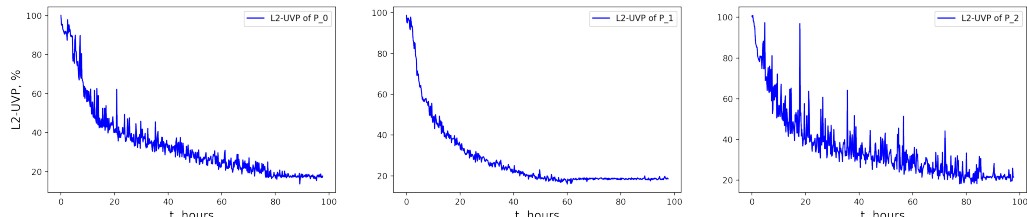

Figure 9: Training curves of $\mathcal{L}$2-UVP vs. time for **OUR** proposed method in Manifold-constrained setup with Style-GAN on Ave, Celeba! dataset. The duration of the training is 100 h (1 GPU V100).

**Convergence at inference.** Since Langevin Dynamics is at the heart of our method for sampling from a conditional plan, it is important to demonstrate the dependence of the inference times/quality on the number of $L$ Langevin steps in Manifold constrained as well as in the Data space setup. We demonstrate Tables 5 and 6, where we show the trade-off between number of Langevin steps $L$ and the obtained quality. To provide the comprehensive analysis, we report FID scores as well as sampling time for different number of steps $L$. Expectedly, inference time linearly depends on $L$ both setups.

Our results testify that (in Manifold constrained setting) our method achieves good quality even with rather small number of Langevin steps, i.e., the computational burden of our approach could be considerably reduced with rather little trade-off in quality.

**Details of the baseline solvers.** For the [55, WIN] solver, we use their pre-trained checkpoints provided by the authors in the above-mentioned repository. Note that the authors of [32, SCWB] do not consider such a high-dimensional setup with RGB images. Hence, to implement their approach in this setting, we follow [55, Appendix B.4].

| FID | | | L | t, sec |
| k=0 | k=1 | k=2 | | |
|---|---|---|---|---|
| 15.8 | 15.3 | 18.3 | 50 | 15 |
| 11.3 | 11.2 | 14.3 | 150 | 45 |
| 8.4 | 8.7 | 10.2 | 250 | 75 |
| 8.3 | 8.2 | 9.9 | 500 | 150 |
| 8.2 | 8.2 | 9.8 | 1000 | 300 |

Table 5: The dependence of running time of ULA on number of Langevin steps during inference mode and the trade-off between the steps and the quality of obtained images in Ave, Celeba! dataset in Manifold constrained setting.

| FID | | | L | t, sec |
| k=0 | k=1 | k=2 | | |
|---|---|---|---|---|
| 125.3 | 122.3 | 187.6 | 10 | 7 |
| 119.4 | 120.0 | 169.7 | 150 | 105 |
| 118.5 | 120.4 | 168.8 | 250 | 175 |
| 118.3 | 120.1 | 168.9 | 500 | 350 |
| 118.8 | 121.2 | 170.2 | 1000 | 700 |

Table 6: The dependence of running time of ULA on number of Langevin steps during inference mode and the trade-off between the steps and the quality of obtained images in Ave, Celeba! dataset in Data space setup.

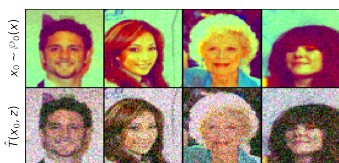 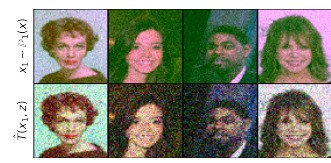 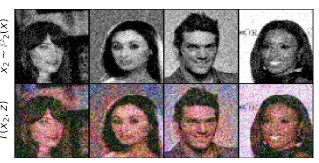

(a) Maps from $\mathbb{P}_1$ to the barycenter. (b) Maps from $\mathbb{P}_2$ to the barycenter. (c) Maps from $\mathbb{P}_3$ to the barycenter.

Figure 10: *Experiment on the Ave, celeba! barycenter dataset.* The plots represent transported inputs $x_k \sim \mathbb{P}_k$ to the barycenter learned by our algorithm in Data space . The true unregularized $\ell^2$ barycenter of $\mathbb{P}_1, \mathbb{P}_2, \mathbb{P}_3$ are situated in figure 4a,4b and 4c correspondingly.

### C.4 Barycenters of Gaussian Distributions

We note that there exist many ways to incorporate the entropic regularization for barycenters [15, Table 1]; these problems do not coincide and yield **different** solutions. For some of them, the ground-truth solutions are known for specific cases, such as the Gaussian case. For example, [75, Theorem 3] examines barycenters for OT regularized with KL divergence. They consider the task

$$\inf_{\mathbb{Q}\in\mathcal{P}(\mathcal{Y})} \sum_{k=1}^{K} \lambda_k \big( \int_{\mathcal{X}\times\mathcal{Y}} \frac{\|x-y\|^2}{2} d\pi_k(x,y) + \epsilon \mathrm{KL}(\pi_k \| \mathbb{P}_k \times \mathbb{Q}) \big) =$$

$$\inf_{\mathbb{Q}\in\mathcal{P}(\mathcal{Y})} \sum_{k=1}^{K} \lambda_k \big( \int_{\mathcal{X}\times\mathcal{Y}} \frac{\|x-y\|^2}{2} d\pi_k(x,y) - \epsilon \int_{\mathcal{X}} H(\pi_k(\cdot|x)) d\mathbb{P}_k(x) + \epsilon H(\mathbb{Q}) \big) =$$

$$\inf_{\mathbb{Q}\in\mathcal{P}(\mathcal{Y})} \bigg\{ \underbrace{\sum_{k=1}^{K} \lambda_k \mathrm{EOT}_{\ell^2,\epsilon}(\mathbb{P}_k, \mathbb{Q})}_{=\text{ our objective inside (5)}} + \epsilon H(\mathbb{Q}) \bigg\}.$$

This problem differs from our objective (5) with $c(x,y) = \frac{1}{2}\|x-y\|^2$ by the non-constant $\mathbb{Q}$-**dependent** term $\epsilon H(\mathbb{Q})$; this problem yields a different solution. The difference with the majority of other mentioned approaches can be shown in the same way. In particular, [67] tackles the barycenter for inner product Gromov-Wasserstein problem with entropic regularization, which is not relevant for us. In turn, the authors of [49] demonstrate significant progress towards GT results for barycenter problem with our considered regularization, but only for univariate (1D) case. To our knowledge, the general Gaussian ground-truth solution for our problem setup (5) is not yet known, although some of its properties seem to be established [25].

Still, when $\epsilon \approx 0$, our entropy-regularized barycenter is expected to be close to the unregularized one ($\epsilon = 0$). In the Gaussian case, it is known that the unregularized OT barycenter for $c_k(x,y) = \frac{1}{2}\|x-y\|^2$ is itself Gaussian and can be computed using the well-celebrated fixed point iterations of [3, Eq. (19)]. This gives us an opportunity to compare our results with the ground-truth unregularized barycenter in the Gaussian case. As the **baseline**, we include the results of [55, WIN] solver which learns the unregularized barycenter ($\epsilon = 0$).

We consider 3 Gaussian distributions $\mathbb{P}_1, \mathbb{P}_2, \mathbb{P}_3$ in dimensions $D = 2, 4, 8, 16, 64$ and compute the approximate EOT barycenters $\pi_k^{\widehat{f}_k}$ for $\epsilon = 0.01, 1$ w.r.t. weights $(\lambda_1, \lambda_2, \lambda_3) = (\frac{1}{4}, \frac{1}{4}, \frac{1}{2})$ with our solver. To initialize these distributions, we follow the strategy of [55, Appendix C.2]. The ground truth unregularized barycenter $\mathbb{Q}^*$ is estimated via the above-mentioned iterative procedure. We use the code from WIN repository available via the link mentioned in Appendix C.2. To assess the WIN solver, we use the unexplained variance percentage metrics defined as $\mathcal{L}_2\text{-UVP}(\hat{T}) = 100 \cdot [\|\hat{T} - T^*\|]_{\mathbb{P}}^2$ where $T^*$ denotes the optimal transport map $T^*$, see [54, §5.1]. Since our solver computes EOT plans but not maps, we evaluate the barycentric projections of the learned plans, i.e., $\overline{\widehat{T}}_k(x) = \int_{\mathcal{Y}} y \pi_k^{\widehat{f}_k}(y|x)$, and calculate $\mathcal{L}_2\text{-UVP}(\overline{\widehat{T}}_k, T_k^*)$. We evaluate this metric using $10^4$ samples. To estimate the barycentric projection in our solver, we use $10^3$ samples $y \sim \pi_k^{\widehat{f}_k}(y|x_k)$ for each $x_k$. To keep the table with the results simple, in each case we report the average of this metric for $k = 1, 2, 3$ w.r.t. the weights $\lambda_k$.

| Dim / Method | Ours ($\epsilon = 1$) | Ours ($\epsilon = 0.01$) | [55, WIN] |
|---|---|---|---|
| 2 | 1.12 | **0.02** | 0.03 |
| 4 | 1.6 | **0.05** | 0.08 |
| 8 | 1.85 | **0.06** | 0.13 |
| 16 | 1.32 | **0.09** | 0.25 |
| 64 | 1.83 | 0.84 | **0.75** |

Table 7: $\mathcal{L}_2$-UVP for our method with $\epsilon = 0.01, 1$ and WIN, $D = 2, 4, 8, 16, 64$.

We see that for small $\epsilon = 0.01$ and dimension up to $D = 16$, our algorithm gives the results even better than WIN solver designed specifically for the unregularized case ($\epsilon = 0$). As was expected, larger $\epsilon = 1$ leads to the increased bias in the solutions of our algorithm and $\mathcal{L}_2$-UVP metric increases.

**Effect of the batch size.** In order to test how our method is affected by batch size, we conduct an additional empirical study with varying batch size. We follow our Gaussian experimental setup from above and pick $(D, \epsilon) = (2, 0.1)$, $(D, \epsilon) = (16, 0.01)$. As the batch sizes, we consider $2, 4, 8, 16, 32, 64, 128, 256, 512, 1024$. As we can see in Table 9, the quality of the recovered plans $\pi^{f_k}$ between reference and barycenter distributions naturally grows with the batch size. In all our other experiments, we typically choose the batch size to be a reasonable number which, on the one hand, allows achieving sufficient quality and, on the other hand, provides reasonable computational burden. In the image experiments, we use batch size $\leq 128$ as it already provides reasonable performance. Going beyond that is challenging due to the computational restrictions.

**Effect of sampling steps number at training.** We conducted an ablation study testing how our method is affected by chosen number of discretized Langevin dynamic steps ($L$ from §4.2) at training, the results are presented in Figure 11. In this experiment, we try to learn the barycenter of Gaussian distributions; we pick the dimensionality $D = 64$ (which is the highest-dimensional case among the considered); $\epsilon = 10^{-1}$. As we can see, performance drops when the number of steps is insufficient. Overall, in all our experiments, the number of Langevin steps is chosen to achieve reasonable qualitative/quantitative results.

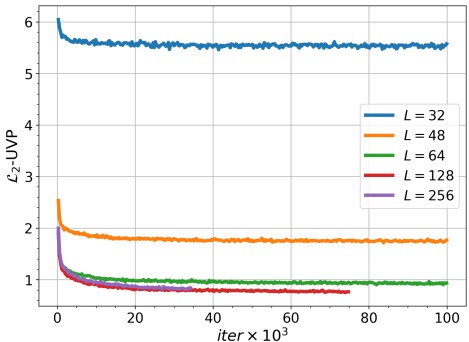

Figure 11: The training curves of $\mathcal{L}_2$-UVP vs. iterations for **OUR** proposed method for the barycenter of Gaussian distributions depending on number of Langevin steps $L$.

## C.5  Single-cell experiment

We consider the problem of predicting the interpolation between single cell populations at multiple timepoints from [107]. The objective is to interpolate the distribution of cell population at any intermediate point in time, call it $t_i$, given the cell population distributions at past and future time-points $t_{i+1}$ and $t_{i-1}$. Since this is an interpolation problem, it is natural to expect that the intermediate population is a (entropy-regularized) barycentric average (with $\ell^2$ cost) of both the population distributions available at the nearest preceding and future times. We leverage the data pre-processing and metrics from paper [56], wherein the authors provide a complete

| Solver/DIM | 50 | 100 | 1000 |
|---|---|---|---|
| **OUR** | **2.32** $\pm$ 0.12 | 2.26 $\pm$ 0.09 | 1.34 $\pm$ 0.12 |
| LightSB-M [2] | 2.33 $\pm$ 0.09 | **2.172** $\pm$ 0.08 | **1.33** $\pm$ 0.05 |
| SFM-sink [3] | 2.66 $\pm$ 0.18 | 2.52 $\pm$ 0.17 | 1.38 $\pm$ 0.05 |
| EGNOT [1] | 2.39 $\pm$ 0.06 | 2.32 $\pm$ 0.15 | 1.46 $\pm$ 0.20 |

Table 8: Energy distance (averaged for two setups and 5 random seeds) on the MSCI dataset along with 95%-confidence interval ($\pm$-intervals). The best solver according to the mean value is **bolded**.

| $(D, \epsilon)$ | Batch size | 2 | 4 | 8 | 16 | 32 | 64 | 128 | 256 | 512 | 1024 |
|---|---|---|---|---|---|---|---|---|---|---|---|
| (2,0.1) | $\mathcal{L}_2$-UVP $\downarrow$ | 0.20 | 0.14 | 0.07 | 0.05 | 0.05 | 0.04 | 0.02 | 0.03 | 0.02 | 0.02 |
| (16,0.01) | | 0.56 | 0.40 | 0.37 | 0.30 | 0.25 | 0.22 | 0.20 | 0.17 | 0.15 | 0.13 |

Table 9: Influence of the batch size on the performance of our approach in the case of computing the barycenters of Gaussian distributions.

notebook with the code to launch the setup similar to [107]. There are 3 experimental settings with dimensions $D = 50, 100, 1000$, each setting contains two setups: predicting day 3 given days 2 and 4 and predicting day 4 given days 3 and 7. The metric is MMD; see [107] or Section 5.3 from [56] for additional experimental details. We report the result in Table 8 where we find that in most cases, our general-purpose entropic barycenter approach nearly matches the performance of leading baselines. This underscores the scope of problems in which our barycentric optimal transport technology can act as a viable foundation model, directly out-of-the-box.

# D    Alternative EBM training procedure

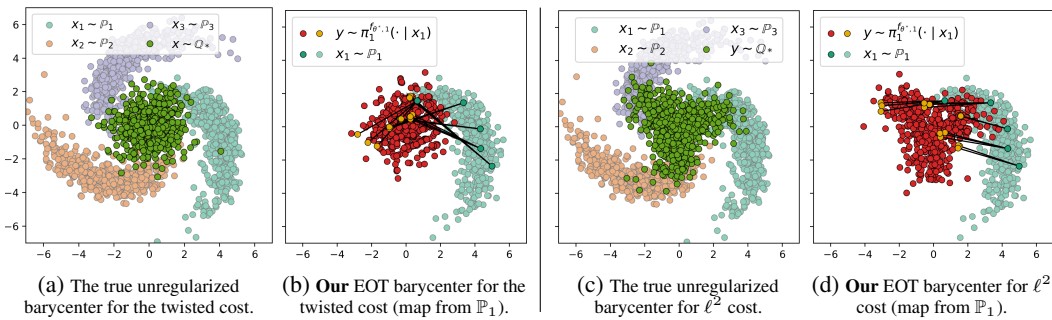

(a) The true unregularized barycenter for the twisted cost.

(b) **Our** EOT barycenter for the twisted cost (map from $\mathbb{P}_1$).

(c) The true unregularized barycenter for $\ell^2$ cost.

(d) **Our** EOT barycenter for $\ell^2$ cost (map from $\mathbb{P}_1$).

Figure 12: *2D twister example. **Trained with importance sampling***: The true barycenter of 3 comets vs. the one computed by our solver with $\epsilon = 10^{-2}$. Two costs $c_k$ are considered: the twisted cost (12a, 12b) and $\ell^2$ (12c, 12d). We employ the *simulation-free* importance sampling procedure for training.

In this section, we describe an alternative **simulation-free** training procedure for learning EOT barycenter distribution via our proposed methodology. The key challenge behind our approach is to estimate the gradient of the dual objective (4.3). To overcome the difficulty, in the main part of our manuscript, we utilize MCMC sampling from conditional distributions $\mu_{x_k}^{f_{\theta,k}}$ and estimate the loss with Monte-Carlo. Here we discuss a potential alternative approach based on **importance sampling** (IS) [105]. That is, we evaluate the internal integral over $\mathcal{Y}$ in (4.3):

$$\mathcal{I}(x_k) \stackrel{\text{def}}{=} \int_{\mathcal{Y}} \left[ \frac{\partial}{\partial \theta} f_{\theta,k}(y) \right] \mathrm{d}\mu_{x_k}^{f_{\theta,k}}(y) \tag{42}$$

with help of an auxiliary proposal (continuous) distribution accessible by samples with the known density $q(y)$. Let $Y^q = \{y_1^q, \dots, y_P^q\}$ be a sample from $q(y)$. Define the weights:

$$\omega_k(x_k, y_p^q) \stackrel{\text{def}}{=} \exp\left( \frac{f_{\theta,k}(y_p^q) - c(x_k, y_p^q)}{\epsilon} \right) q(y_p^q).$$

Then (42) permits the following stochastic estimate:

$$\mathcal{I}(x_k) \approx \frac{\sum_{p=1}^{P} \left[ \frac{\partial}{\partial \theta} f_{\theta,k}(y) \right] \omega_k(x_k, y_p^q)}{\sum_{p=1}^{P} \omega_k(x_k, y_p^q)}. \tag{43}$$

**Experimental illustration.** To demonstrate the applicability of IS-based training procedure to our barycenter setting, we conduct the experiment following our **2D Twister** setup, see §5.1. We employ zero-mean $16I$-covariance Gaussian distribution as $q$ and pick the batch size $P = 1024$. Our results are shown in Figure 12. As we can see, the alternative training procedure yields similar results as Figure2 but converges faster ($\approx 1$ min. VS $\approx 18$ min. of the original MCMC-based training).

**Concluding remarks.** We note that IS-based methods requires accurate selection of the proposal distribution $q$ to reduce the variance of the estimator [105]. It may be challenging in real-world scenarios. We leave the detailed study of more advanced IS approaches in the context of energy-based models and their applicability to our EOT barycentric setup to follow-up research.

