# OpenReview forum: "Energy-Guided Continuous Entropic Barycenter Estimation for General Costs"
_NeurIPS.cc/2024/Conference — NeurIPS 2024 spotlight_

### Official Review · Reviewer_LoLb · 2024-07-10

**Soundness:** 3
**Presentation:** 2
**Contribution:** 3
**Rating:** 7
**Confidence:** 4

**Summary:**

This paper deals with the question of estimating Wasserstein(-like) barycenters in the _continuous_ case based on _samples_. That is, one actually observes (iid) samples $X^k$ made of $N_k$ points from (unknown) probability distributions $\mathbb{P}_1,\dots,\mathbb{P}_K$, and the goal is to estimate $\mathbb{Q}^*$ the barycenters of the $(\mathbb{P}_k)_k$ (and not simply the discrete barycenter of the $X^k$).

The idea developed in this work is to maximize a loss (arising from a dual problem of an _entropy-regularized_ OT formulation) $L(f_1,\dots,f_K)$ over $K$ potential $(f_k)_k$ (themselves parametrized by neural networks). A given $f_k$ induces a measure $\pi^{f_k}$ whose first marginal is $\mathbb{P}_k$, and the second marginal is some $\mathbb{Q}_k$ that aims at approximating $\mathbb{Q}^*$.
Of interest, $\pi^{f_k}$ admits a disintegration and its second marginal can be sampled by the following scheme :
- sample $x_k$ from $\mathbb{P}_k$ (i.e. take an observation in the training set),
- Then sample $y \sim \mu_{x_k}^{f_k} \propto e^{-V}$ with potential $V(y) \coloneqq c(x_k, y) - f_k(y)$, typically using a Langevin-type approach.
The objective value $L$ itself may not be accessible in close form (because the normalization constant of $\mu_{x_k}^{f_k}$ is unknown), but its gradient---as usual in score-based problem---is accessible and optimization is thus possible though a practical algorithm.

From a theoretical standpoint, authors prove that the error gap $(\max L) - L(f_1,\dots,f_K)$ controls the distances $\mathrm{KL}(\mathbb{Q}^* | \mathbb{Q_k})$, i.e. a small error gap ensures that the second marginals of the $\mathbb{Q_k}$ should be close to the target $\mathbb{Q}^*$. They also establish a statistical consistency result, in that if the number of observations $(N_k)_k$ goes to $\infty$ (for all $k$), the estimation error of their model goes to zero.

The approach is showcases on toy and images datasets.

**Strengths:**

First, the paper is well-tailored for NeurIPS in my opinion: it tackles a problem already studied in the ML community, propose a clear methodology that yields a practical algorithm and that is supported by reasonable theoretical results, and conduct both toy and demanding experiments. The main paper is mostly self-contained, while the appendix provides insightful complementary comments.

The article provides a fairly good report of sota models/related works.

Moreover, the approach can handle general cost functions (and not only the widely used squared Euclidean cost), and this advantage is nicely showcased in the experiment by enabling to _parametrize_ the barycenter (e.g. by leveraging a prior) on a latent space $Z$, i.e. the cost becomes $c(x,z) = |x - G(z)|^2$ for instance. This perspective is satisfactory as it opens the way for interesting applications of OT-based barycenters.

**Weaknesses:**

1. The main weakness (and for which I unfortunately do not have a specific solution) of the approach is the difficulty to properly evaluate it (note: this is not specific to the method but rather to the problem considered), due to:
- the actual (regularized, or not) barycenter $\mathbb{Q}^*$ being usually unknown,
- the value of the loss $L$ (a fortiori the optimal value $L^*$ and the error gap) cannot be computed,
- even if we could compute it, it is natural to run experiments with a small regularization parameter $\epsilon$, but in this regime the approximation bound provided by Theorem becomes useless.
It thus makes very hard to "monitor" the algorithm and to have a practical clue on whether its output can be trusted or not. This does not invalidate the methodology in my opinion, but remains an important weakness that must be stressed.

2. In terms of clarity, while I consider that the "methodology" part of the work (section 4) is reasonably well-written, the experimental section is much harder to follow in my opinion: I do not exactly understand if "$\ell^2$-barycenters" is supposed to mean "Wasserstein barycenters for the quadratic cost" or "barycenter between images seen as vector (standard means)". Explicitly, lines 317--320 claim that those are "piwel-wise average" (so, in my understanding, standard means) "coming from $\ell^2$ OT plan" (so, in my understanding, geodesics/McCann interpolation; from which you have access to the Wasserstein barycenter).

3. This work takes many ideas from [77] (energy-based optimization, etc.), which to some extend diminishes its originality. This is discussed in lines 813--819. I believe that this should belong to the main paper and be slightly extended, explaining more precisely what are _the theoretical and practical difficulties_ that had to be overcome to go from [77] to the actual work (the discussion refers to "multiple marginals, optimization with an unfixed marginal $\mathbb{Q}$", and I agree with that, but in some sense the proposed approach seems to adapt so naturally (which is good!) from [77] that I do not see "what could have gone wrong").

**Questions:**

1. The approach relies on a Langevin (or other sampling) scheme with potential energy given by $f_k - c(x_k, \dot)$. Given that $f_k$ is encoded by a neural network which is typically neither convex not smooth in the space variable $y$ (nor in the parameters $\theta$ but that does not matter here), it's dubious to assume that the Langevin scheme converge toward the target distribution. This is only very briefly mentioned in lines 363-364 and in line 1051 and I believe this should be discussed more extensively (i.e. did you notice it during your experiments? how impactful is it?)

2. While the choice the entropic regularization ($H(\pi)$, $\mathrm{KL}(\pi | \mathbb{P} \otimes \mathbb{Q})$...) is not crucial _when the marginal are fixed_, it starts to play a role when optimizing over a given marginal, as in the barycenter problem. This work consider the Schrodinger barycenter (using the terminology of Chizat [15, Table 1]) as discussed in appendix B.3---which also discusses the importance of this choice from a practical viewpoint. However, this modelling induces a "blurring" bias which is never mentioned in the work as far as I can tell. Can you briefly elaborate on this? For instance, if the input distribution are actually given by $\delta_{x_k}$, what would be the output of the algorithm?

3. As far as I can tell, the barycenter that is targeted here similar (if not exactly the same) as the $\alpha_{\mathrm{OT}^\mathcal{U}}$ of Janati et al., _Debiased Sinkhorn barycenters_ (note: this work should be cited/discussed at some point). is that correct? If so, note that in that case an  _closed form_ expression for the entropic barycenter of (univariate) Gaussian distributions is accessible (showcasing formally the blurring effect); and section C.4. may be revisited under this lens.

**Limitations:**

Yes, the authors discuss the limitations of their work in a convincing way (in the appendix; ).

---

> ### Author Rebuttal · Authors · 2024-08-07
>
> Dear reviewer, thank you for spending time reviewing our paper. Your kind words that our paper well suits NeurIPS are encouraging for us. Below you can find the answers to your comments and questions.
>
> **(W1) Difficulties with proper evaluation.**
>
> We agree that the out-of-the-box evaluation is typically unavailable for the problem we aim to tackle (entropic barycenter). This is due to the inherent *complexity* of the barycenter problem. In particular, there seem to be no easy-to-calculate statistics to monitor to control the quality of the recovered barycenter (like the loss gap $L - L^*$). Surprisingly, we were able to derive a solution to the notoriously challenging entropic barycenter problem. On the other hand, we did our best to validate our methodology as accurately as possible. In particular, we consider several experimental setups where the unregularized barycenter is **known**: (a) Toy 2D Twister (Section 5.1), (b) Ave Celeba experiment (Section 5.3), Gaussian case (Appendix C.4). These setups allow us quantitatively/qualitatively estimate the recovered EOT barycenter.
>
> *A comment on small $\epsilon$ regime.* From the theoretical point of view (approximation bounds etc.), the regime $\epsilon \rightarrow 0$ is indeed not so favourable. However, in terms of method evaluation, it is the regime that allows more-or-less accurate ways to "monitor" the quality. This is thanks to the aforementioned practical setups, where the true *unregularized* barycenters are known, and we can directly compare them with the generated ones. Practically speaking, in our paper, we consider rather small values of $\epsilon$ (up to $10^{-4}$), and our approach treats such choices.
>
> **(W2) Questions about the experimental section and $\ell^2$ barycenters.**
>
> Your understanding is correct: when we consider $\ell^{2}$ barycenters of images, the $\ell^{2}$ between two particular images is computed by flattening each image into a vector (e.g., $32\times 32\rightarrow 1024$). In turn, when there are **just two** distributions ($K=2$) of images considered, their $\ell^{2}$ barycenter is indeed a certain McCahn interpolant. However, one needs to recover somehow the true $\ell^{2}$ OT map (plan) between the distributions to access the barycenter through this interpolant.
>
> **(W3) Our work vs. [77]. Theoretical and practical efforts upon [77] which are required to end up with our approach.**
>
> Please find the answer to your comment in the general answer to all the reviewers.
>
> **(Q1) Non-convexity of the energy potentials. Extending the discussion and practical observations.**
>
> The problem with the non-convex energy function and its effect on MCMC is a hallmark of energy-based methods (and even score-based approaches). The non-convexity may cause some MCMC chains to become trapped by local minima of the energy landscape, making the corresponding samples somewhat different from the expected samples from $\pi^f_k(y \vert x_k)$. By taking a closer look at our results, e.g., Figure 7 (conditional samples from MNIST 0/1 barycenter), one may distinguish that some generated pictures "fall out of the line", i.e., noticeably differ from the others. We think this phenomenon is inherently caused by the non-convexity. We underline that non-convexity does not stop EBMs from successfully modelling high-dimensional distributions (see, e.g., references [34], [112] from our paper), which makes our methodology practically appealing. In the revised version of our manuscript, we will expand the discussion on the non-convexity.
>
> **(Q2) The type of the considered Entropic Barycenter problem. "Blurring" effect. Entropic barycenter of $\delta$-distributions.**
>
> Please note that we directly mention the blurring bias in our experiment with MNIST01 in data space (section 5.2, Figure 5 and lines 323-325) in our paper. However, we do not focus on it too much because in the manifold-constrained setups (MNIST01 in Section 5.2, Ave, Celeba in Section 5.3) this bias in fact disappears. More generally, we believe that manifold-constrained setups are generally more interesting and probably even necessary to obtain meaningful barycenters for downstream practical problems (see Appendix B.2 for a discussion).
>
> Regarding the $\epsilon$-barycenters of $\mathbb{P}\_{k}=\delta_{x_{k}}$, it seems like it is some Gaussian centered at $\sum_{k=1}^{K}\lambda_k x_{k}$ with variance proportional to $\epsilon$ (in the case of $\ell^{2}$ barycenters). Namely, the higher $\epsilon$ is, the more "blurry" (noisy) the $\epsilon$-barycenter is compared to the unregularized one. Following your question, we will add an extended discussion of this aspect to B.3.
>
> **(Q3) Missed citation. (Possible) closed-form expression for entropic barycenters in 1D.**
>
> Thanks for noting, we were not aware of this paper. It indeed seems like their $\alpha_{\text{OT}^{\mathcal{L}}}$ barycenter coincides with our considered Schrodinger barycenter. We will surely include the citation and proper discussion, including their formula for the closed-form barycenter for 1D Gaussians and $\ell^{2}$ cost.
>
> **Concluding remarks.** Kindly let us know if the clarifications provided address your concerns about our work. We're happy to discuss any remaining issues during the discussion phase. Thank you one more time for your efforts.

---

> > ### Comment · Reviewer_LoLb · 2024-08-08
> > **Thanks**
> >
> > Thanks for taking time answering my review and providing clarification / details on the few questions I had. So far, I am satisfied with the work, I'll try to engage in discussion with other reviewers to make my mind clearer.

---

### Official Review · Reviewer_wGnY · 2024-07-11

**Soundness:** 3
**Presentation:** 3
**Contribution:** 3
**Rating:** 6
**Confidence:** 3

**Summary:**

The authors propose a new algorithm to estimate entropic barycenters in this submission.
In particular, leveraging the c-transform based duality of the entropic OT problem, the authors reformulate the entropic barycenter problem as a constrained functional optimization problem, in which a set of dual potential functions are optimized to minimize the energy function corresponding to the c-transform.
The potential functions are parametrized by neural networks so that the constrained functional optimization problem can be implemented as an unconstrained and parametric optimization problem, which can be solved efficiently by the gradient descent associated with MCMC sampling.
Compared to existing OT-based barycenter estimation methods, the proposed algorithm can be applied to general cost functions and can explicitly estimate OT plans.

**Strengths:**

- The motivation is clear, and the derivations of key formulas are clear and easy to follow.
- The theoretical part is solid, which supports the rationality of the proposed method, including its sample complexity, generalization power, and so on.

**Weaknesses:**

- It has been well-known that entropic OT suffers from the sensitivity of the entropic regularizer’s weight. It is not clear to me why the proposed method can support high-dimensional applications. Especially, in Lines 208-212, epsilon impacts a lot on the MCMC sampling step. The robustness test of the proposed method to the regularizer’s weight should be considered in the experimental part.
- Parameterizing continuous potential functions by neural networks means shrinking the feasible space of the problem. How to design the neural networks? Again, the impacts of different NN models and the robustness of the method to the model selection should be considered.
- The proposed algorithm involves MCMC sampling and gradient descent, so the convergence and the computational efficiency of the proposed algorithm should be analyzed quantitatively. In particular, in high-dimensional cases, do we need to sample a lot?

Minors:
- It would be nice if the authors could apply the proposed method in a practical task, e.g., point cloud clustering.
- For the completeness of the main paper, a conclusion section should be added.

**Questions:**

Please see above.

**Limitations:**

Not applicable.

---

> ### Author Rebuttal · Authors · 2024-08-07
>
> Dear reviewer. Thank you for spending time reading and evaluating our work.  We were greatly encouraged by your positive feedback on our theory and the readability of our manuscript.  Below you can find the answers to your questions and comments.
>
> **(W1) The role and impact of entropic regularizer's weight. Robustness test of the method to the regularizer's weight.**
>
> We agree that considering too small $\epsilon$ within our proposed method may cause some difficulties with optimizing our energy-based barycenter objective. The heart of the matter is the answer to:
> "can our approch permit *practically-reasonable* choices for $\epsilon$"?
> We believe that our considered practical use cases (Figure 5, Figure 7, Figure 8) testify that this is the case. The images sampled from our learned conditional plans $\pi^k(y \vert x)$ are of reasonable diversity. We want to emphasize that for Ave Celeba! experiment (manifold constrained) the considered value of $\epsilon$ was $10^{-4}$.
> Also, as per request, we additionally deployed our methodology on MNIST 0/1, manifold setup, with an $\epsilon$ of $10^{-4}$ (in addition to the previously considered value of $\epsilon=10^{2}$).  The results are demonstrated in **Figure 1 in the attached pdf**.
>
> Regarding the high-dimensional applications, our comment is as follows. It seems that learning an (especially entropic) barycenter directly in the image data space is, in most cases, not very meaningful - we will obtain just blurred pixel-wise averages of input images. This is exactly where our StyleGAN trick comes into play. In turn, the latent space of StyleGAN is moderate dimensional and allows learning meaningful barycenters in high-dimensions. And our experimental results demonstrate that our approach successfully tackles this setup.
>
> **(W2) NNs architectures to parameterize the potential functions.**
>
> One of the key insights of our method in practice is the reliance of our approach on already well-established, built-on efficient architectures (and hyper-parameter selections) constructed for EBMs and EOT problems [1]. Since we trust the architectural design and hyperparameter specifications of those authors, we perform no additional find-tuning, using their neural networks as pre-trained foundation models. We think that doing this, in addition, is out of the scope of the current work. Please note that our primal contribution is methodological, not architectural, but has a primarily theoretical intent.
>
> **(W3) Reliance on MCMC and gradient descent. Quantitative analysis of convergence and computational efficiency in, e.g., high-dimensional case.**
>
> We adopt MCMC and (stochastic) gradient descent as the standard tools for Energy-based Models. It is a generic knowledge that these tools allow for generating high-dimensional images (when carefully tuned); see references [34], [112] from our paper.
>
> Following your request, we quantitatively analyze the performance of our method as a function of the number of discretized Langevin dynamic steps, see the charts (Figure 4) in our attached pdf.
> In this experiment, we try to learn the barycenter of Gaussian distributions; which, as explained in Appendix C.4, is known in-closed form for the unregularized case, which gives us access to a theoretical object by which we may compare and validate our computational pipeline. We pick the dimensionality $D = 64$ (which is the highest-dimensional case among the considered). As we can see, performance drops when the number of steps is insufficient. Overall, in all our experiments, the number of Langevin steps is chosen to achieve reasonable qualitative/quantitative results.
>
> **(Minor W4) Practical application.**
>
> For an intriguing practical application, we pick an intriguing Single-Cell Data analysis experiment, where we need to predict the distribution of cell population at some intermediate time point $t_i$ given cell distributions at $t_{i + 1}$ and $t_{i - 1}$. Please see a brief explanation (and the relevant links) of the problem in our General response, our results are in Table 1 in the attached pdf. Our approach achieves competitive performance.
>
> **(Minor W5) Call for conclusion section.**
>
> Section E contains our conclusions and summary. Currently, this section is located in the Appendix. However, following your suggestion, we will move it to the main part of our manuscript in the final version.
>
> **Concluding remarks.** Kindly let us know if the clarifications provided address your concerns about our work. We're happy to discuss any remaining issues during the discussion phase. If you find our responses satisfactory, we would appreciate it if you could consider raising your score.
>
> **References**
>
> [1] Mokrov et. al., Energy-guided entropic neural optimal transport. ICLR'2024.

---

### Official Review · Reviewer_unyK · 2024-07-12

**Soundness:** 3
**Presentation:** 2
**Contribution:** 2
**Rating:** 6
**Confidence:** 3

**Summary:**

The authors focus on the Wasserstein barycenter problem; an average between distributions given a Wasserstein cost. Specifically, they consider arbitrary cost functions to define the Wasserstein distance and they approximate the continuous barycenters. They further relate their method with energy based models, and show experiments and synthetic and image datasets.

**Strengths:**

The method is theoretically well-founded and has many applications. I particularly found interesting that it can work for multiple different ground cost functions.

**Weaknesses:**

- The background section may have too many details, which undermine readability and makes it harder to understand the main assumptions and challenges. I think some of it could be moved to the appendix. In particular, it is not clear to me that all definitions are useful, as some of them do not seem to be used in the subsequent sections (e.g. $\hat{\mathbb{P}}$). I believe the authors could improve readability by keeping only the main assumptions and definitions (hence notations) in the main body of the text, and moving other results, albeit very interesting, to the appendix.

- It is a bit unclear how this work is different from the work [1]. Could the authors elaborate on this? Further, I believe it is worth comparing with [2], as it is now published at ICML 24.

- Most of the experimental results are qualitative, which makes it difficult to attest if the method improves on existing ones. I understand that for most distributions the ground truth barycenter is unknown, but it could be approximated by the unregularized barycenter. Otherwise, the barycenter between two distributions relates to the McCann interpolation, a potential task could be to predict the interpolation between distributions at multiple timepoints (e.g. Tab.4, Tab.5 in [3]). A task on treatment effect using barycenters could also be used (similar to [4]).

[1] Petr Mokrov, Alexander Korotin, Alexander Kolesov, Nikita Gushchin, and Evgeny Burnaev. Energy-guided entropic neural optimal transport. In The Twelfth International Conference on Learning Representations, 2024.

[2] Alexander Kolesov, Petr Mokrov, Igor Udovichenko, Milena Gazdieva, Gudmund Pammer, Evgeny Burnaev, and Alexander Korotin. Estimating barycenters of distributions with neural optimal transport. ICML, 2024.

[3] Tong, A., Malkin, N., Fatras, K., Atanackovic, L., Zhang, Y., Huguet, G., ... & Bengio, Y. (2023). Simulation-free schr\" odinger bridges via score and flow matching. arXiv preprint arXiv:2307.03672.

[4] Huguet, G., Tong, A., Zapatero, M. R., Tape, C. J., Wolf, G., & Krishnaswamy, S. (2023, September). Geodesic Sinkhorn for fast and accurate optimal transport on manifolds. In 2023 IEEE 33rd International Workshop on Machine Learning for Signal Processing (MLSP) (pp. 1-6). IEEE.

**Questions:**

- What is $\mathcal{B}(\mathbb{Q})$ in eq. 8 ?
- Section 2.2 could be shortened, I believe we only really need to know that the barycenter is unique if it exists.

## Minor comments and potential typos
- Maybe a missing word on line 86 "has particular".
- "falls of the scope" $\to$ falls within the scope of the paper.
- The authors should refer to the proofs of the theorems in the main body of the text, even if the proof is in appendix.
- In Tab.2, could you add the standard deviation and the names of the other two solvers.

**Limitations:**

The authors address the limitations of their work in the manuscript.

---

> ### Author Rebuttal · Authors · 2024-08-07
>
> Dear reviewer. Thank you for your thoughtful review. We are pleased that you noted the theoretical and practical strengths of our work. Below we answer your comments and questions.
>
> **(W1)+(Q2). Too many details in the background section. Improving the readability by moving some secondary assumptions/definitions to the Appendix. Shortening Section 2.2.**
>
> Following the reviewer's request, in the revised version of our manuscript, we will shorten the Background section by leaving only vital definitions/assumptions in the main part. In particular, we will remove/move to the appendix Eq. (1) (definition of $\text{EOT}^{(2)}$). Indeed, only Eq. (2) actually needed. Correspondingly, Eq. (7) will also disappear from the main part. Following your request, we will also shorten the explanations after Eq. (8) (Section 2.2) and eliminate introducing empirical measure $\widehat{\mathbb{P}}$, line 121.
>
> **(W2. Part 1) The difference between this work and [1].**
>
> Please, take a look at our General response (1) to all the reviewers, where we get into details of the connection with [1] and our novelty over [1].
>
> **(W2. Part 2) Comparison with [2].**
>
> We highlight that the authors of [2] acknowledge that *their work is significantly based on our paper*, see the Contribution subsection in [2]. Interestingly, they also seem to conduct a comparison of their approach with ours on Ave Celeba! barycenter benchmark. According to their results, on the manifold-constrained data setup, our method achieves FID scores $\approx 9$, while their approach yields a much bigger number $\approx 30$. Therefore, our method demonstrates a significant quantitative boost. This is a clear indicator of the strength of our energy-guided methodology.
>
> **(W3) "Most of the experimental results are qualitative, which makes it difficult to attest if the method improves on existing ones." Call for more experiments with quantitative outcomes.**
>
> Following your kind suggestion, we conducted a novel experiment with predicting the interpolation between distributions at multiple timepoints following the setup identical to [3]. Please refer to the general answer to all the reviewers. According to the results, our method works on par with competitive methods.
>
> For completeness, we also note that we have a purely quantitative experiment in the Gaussian case in Appendix C.4.
>
> **(Q1) "What is $\mathcal{B}(\mathbb{Q})$ in eq. 8?"**
>
> The term $\mathcal{B}(\mathbb{Q})$ is defined directly in equation (8), i.e., $\mathcal{B}(\mathbb{Q}):= \sum\limits_{k = 1}^{K} \lambda_k \text{EOT}_{c_k, \varepsilon}(\mathbb{P}_k, \mathbb{Q}).$ We agree that this makes (8) to be a bit overcomplicated, which was done due to space constraints. Sorry for this. In the final version, we will have one extra page and we will separately define $\mathcal{B}(\mathbb{Q})$ or even get rid of this notion completely.
>
> **Minor comments and typos**
>
> Thank you for pointing them out! We will fix them in the revised version of the manuscript. The FIDs for our method with std. deviations (running the inference with different seeds) look as:
>
> | k=1 | k=1  | k=2  |
> |------|------|------|
> | 8.4 (0.3)  | 8.7 (0.3)  | 10.2 (0.7) |
>
> For the competitors, we grab the results from WIN paper where no std. deviations are provided.
>
> **Concluding remarks.** Kindly let us know if the clarifications provided address your concerns about our work. We're happy to discuss any remaining issues during the discussion phase. If you find our responses satisfactory, we would appreciate it if you could consider raising your score.
>
> **References**
>
> [1] Mokrov et. al., Energy-guided entropic neural optimal transport. ICLR'2024.
>
> [2] Kolesov et. al., Estimating barycenters of distributions with neural optimal transport. ICML'2024.
>
> [3] Tong et. al. Simulation-free Schrödinger bridges via score and flow matching. AISTATS'2024

---

> > ### Comment · Reviewer_unyK · 2024-08-08
> > **Thanks**
> >
> > Thank you for your response, I appreciate the clarifications and additional experiments. It is now clearer to me how this work differs from [1].

---

### Official Review · Reviewer_kw6j · 2024-07-12

**Soundness:** 3
**Presentation:** 3
**Contribution:** 3
**Rating:** 6
**Confidence:** 4

**Summary:**

The paper proposes a methodology to approximate (regularised) barycenters of continuous-support distributions for arbitrary transport costs. To do so, the authors combine strategies and results from weak OT and transport map approximation via neural networks. The proposal is well connected with the existing (ever-growing) literature on barycentric approximation and provides a set of theoretical results as well as numerical examples of real-world data.

**Strengths:**

The paper is clearly written and well-positioned in the literature. The contribution features theoretical guarantees with their respective proofs and multiple experiments with different toy and benchmark datasets.

**Weaknesses:**

- The use of the Style GAN seems to be pivotal in the experimental validation, yet it is only explained briefly while reporting the experiments.

- There is little mention of the computational complexity (either conceptually or empirical) related to the method and how it can affect its adoption by others.

- There are some formatting issues:
  - line 255: _Substituting (13) or (14) to (13) ..._
  - lines 301-302 have spacing problems
  - the top of fig 5 is cropped
  - many formatting problems with the references

**Questions:**

Please refer to my comments on computational complexity and the role of the Style GAN.

In particular, after reading the paper, it seems to me that the use of the Style GAN was adopted during the final stages of the paper production, therefore, there's only brief mentions of it in the experiments but without the required /deserved attention, despite being a fundamental part of the success of the experiments.

Please elaborate on this point and, if possible, include it in the paper

**Limitations:**

Though the paper features a section called limitations, the authors only state that the one limitation of their method was alleviated by using Style GANs. A deeper identification and discussion of the limitations of the proposed method is lacking

---

> ### Author Rebuttal · Authors · 2024-08-07
>
> Dear reviewer. Thank you for your efforts in reviewing our paper and for your valuable comments. We are delighted that you appreciated both the theoretical and practical contributions of our paper. Below you can find the answers to your questions and comments:
>
> **(W1) "The use of the Style GAN seems to be pivotal in the experimental validation, yet it is only explained briefly while reporting the experiments." Proper attention to the idea with StyleGAN manifold.**
>
> In the revised version of our manuscript, we will devote a specific subsection to the idea with StyleGAN manifold. We will explain the technique itself and clarify why it is significant and valuable in the context of entropic barycenters. Our reasoning here is approximately as follows:
>
> * At first, solving the entropic barycenter problem directly in the image data space leads to noisy images, see, e.g., our MNIST 0/1 experiment. Such "deterioration" follows from the theory - in fact, the Schrödinger barycenter problem we consider (see Appendix B.3) is known to have blurring bias, see work [15] from the references in our manuscript. Our practical reliance on MCMC also makes it obvious that the generated images from the barycenter are noisy.
> * Secondly, solving barycenter problems in image data space **is not very practical** (for standard costs $c_k$ like $\ell^2$ or around). Indeed, our paper (MNIST 0/1 experiment in data space) and previous research (see, e.g., [1, Figure 7]) witness that $\ell^2$ barycenter of image distributions $A$, $B$, $C$ etc.\ is a bizarre distribution which includes "averaged" images of $a \in A$, $b \in B$, $c \in C$ etc. So, we think that direct manipulation with image data space is more about benchmarking barycenter solvers rather than real-world applicability.
> * In contrast, operating with images with the assistance of our proposed StyleGAN-inspired manifold technique seems to be **much more practically appealing**, see the examples of potential practical applications in our Appendix B.2.
>
> **(W2) Computational complexity of the method.**
>
> As per request, we report the (approximate) time for training and inference (batch size = $128$) of our method on different experimental setups, see the table below (the hardware is a single V100 gpu):
>
> |Phase\Experiment   | Toy $2D$  | Gaussians  | Sphere   | MNIST(0/1) (data)  | MNIST(0/1) (manifold)  |  Ave Celeba (data) |  Ave Celeba (manifold) |
> |---|---|---|---|---|---|---|---|
> | Training  | 3h   | 6h  | 3h  | 20h  | 10h  | 60h  | 60h  |
> |  Inference  | 20s   | 40s  | 20s  | 1m  | 1m   | 2m  | 2m  |
>
> For Ave Celeba experiment, we additionally report the computational complexity of the competitors (partially grabbed from the original papers, partially obtained through our own evaluations):
>
> | Phase\Experiment   | WIN  | SCWB  |
> |---|---|---|
> | Training  | 160h  | 200h  |
> |  Inference |  10s | 10s  |
>
> As we can see, all the methods in this experiment require a comparable amount of time for training. The inference with our approach is expectedly costlier than competitors due to the reliance on MCMC. In the revised version of our paper, we will add the information about the computational complexity.
>
> **(L1) Identification and discussion of the limitations apart those alleviated by StyleGAN.**
>
> In Appendix E we have a paragraph called "**Methodological** limitations" which discusses the limitations not related to StyleGAN. When preparing the manuscript, we accidentally leave the word "limitations" in the "**Methodological** limitations" to be normal, not bold. It is our bug, sorry for this. Will be fixed.
>
> **Formatting issues.**
>
> Thank you for pointing them out! We will go through the text carefully and do our best to fix the formatting problems in the final revision.
>
> **Concluding remarks.** Kindly let us know if the clarifications provided address your concerns about our work. We're happy to discuss any remaining issues during the discussion phase. If you find our responses satisfactory, we would appreciate it if you could consider raising your score.

---

> > ### Comment · Reviewer_kw6j · 2024-08-09
> > **Acknowledge of rebuttal**
> >
> > Dear Authors,
> >
> > many thanks for the reply. The contribution of the paper is much clearer to me know. I appreciate the clarification about computational complexity.

---

### Official Review · Reviewer_xS43 · 2024-07-24

**Soundness:** 4
**Presentation:** 3
**Contribution:** 3
**Rating:** 7
**Confidence:** 4

**Summary:**

This paper presents a new entropically regularized algorithm for identifying the Wasserstein barycenter (distribution on average closest to the reference distributions as measured in Wasserstein distance) among K distributions.

The proposed algorithm relies on a dual form of the entropic OT. Namely the entropic OT can be formulated as a supremum over the potential f and its entropic c-transform, the latter of which is defined by a measure whose partition function will be exploited in a learning algorithm defined by the objective function given in (12).

The authors proceed to provide a theoretical analysis of the proposed algorithm, providing KL and universal approximation bounds.

They then follow with a number of experimental demonstrations of the method.

**Strengths:**

The paper nicely exposits how arrive at the primary objective function for the entropic algorithm, and along the way provides good intuition for where the EBM picture emerges (labeling of the "partition function" (5), etc...) and its multimarginal generalization.

The experiments seem pretty thorough, though I must say it is quite hard to test how well a Wasserstein barycenter algo works, as we have few ground truth examples to compare to, and the high-dimensional examples are always quite subjective.

**Weaknesses:**

- Please correct me if I'm wrong, but the discussion about convergence bounds seems a bit distracting because this is not in some sense a measurable convergence diagnostic (you do not know necessarily what $\mathcal L^*$ in the second half of Prop 4.4). While it is good to show that the loss controls these KLs, Theorem 4.6 to me is what seems distracting unless it actually informs the method. We know NNs are universal function approximators.
- Because multimarginal problems are usually quite hard to benchmark (as we have few known solutions), such discussions would indeed be appealing if they were practically informative, as we also know that the multimarginal problem is np-hard.

Some small remarks:

- Line 208: ULA is surely not the simplest MCMC algorithm.
- personal complaint: the notation feels a bit heavyhanded and can be distracting from the main story of the paper.

**Questions:**

It is often the case in energy-based modeling that evaluating the loss (12) can be very high variance because of the necessity of the MCMC procedure and can have negative effect on the performance of the algorithm. Do the authors notice any such difficulty in doing so? How in general do you see benchmarking performance, which is usually important when an algorithm relies on stochastic gradient descent and a neural network, rather than stricter algorithms usually associated to EOT (like sinkhorn). I don't mean to be critical here, just reflective. The toy distributions example is nice.

---

> ### Author Rebuttal · Authors · 2024-08-07
>
> Dear reviewer, thank you for spending time reviewing our paper and for a positive evaluation of the work. We are happy that you found our work to be interesting and well-presented. Below we answer your questions and comments.
>
> **(W1) Convergence bounds discussions. "[...] it is not [...] a measurable convergence diagnostic". Significance of Theorem 4.6.**
>
> It seems that there is a tiny misunderstanding. The results from Section 4.3 (Prop. 4.4, Theorem 4.5, Theorem 4.6) exhibit the generalization and approximation properties of our proposed solver. Namely, we give a complete (statistical) picture of what happens when we move from real reference distributions $\mathbb{P}_k$ to their empirical counterparts. To the best of our knowledge, the ability to carry out such the analysis is exceptional in the existing neural OT literature, which strengthens the contribution of our work. While our Prop. 4.4 indeed does not give particular figures on how different the true $\pi_k^*$ and recovered $\pi^{\widehat{f}_k}$ plans are, it provides the rates of convergence (w.r.t. number of empirical samples). Indeed, if we assume the approximation error to be zero (note Theorem 4.6), then Theorem 4.5. gives us the desired rates.
>
> While our Theorem 4.6. indeed looks simply like "we have a loss function, and can approximate it arbitrarily good with NNs", this result is not that easy. Indeed, the considered loss $\mathcal{L}$ operates with (a) *congruent* potentials and (b) deals with entropic $C_k$-transforms of the potentials, which complicate the overall analysis. In particular, only specific properties of the entropic $C_k$-transforms allow deriving the desired universal approximation statement, see the proof of the Theorem.
>
> **(W2) Discussions on intrinsic difficulty of multimarginal problems.**
>
> The problem we aim to tackle is indeed significantly complex by nature. The fact that our method theoretically (and practically) recovers entropic barycenters is quite encouraging for us. Having in mind the complexity of the barycenter problem, in our paper we pay special attention to the experimental setups where our method could be evaluated qualitatively or even quantitatively, see, e.g.:  Toy 2D Twister (Section 5.1), Ave Celeba  (Section 5.3), Gaussian case (Appendix C.4).
>
> At the same time, we humbly think the discussions on the NP-hardness of the multimarginal problems are out of the scope of our paper, since our approach is based on parametric models (Neural Networks) learned with gradient descent/ascent, which eliminates such questions coming from the Theory of Algorithms.
>
> **(Q1) Practical aspects of optimizing Energy-based training objectives. Variance of the loss estimate. Reliance on MCMC, stochastic gradient descent, and neural networks.**
>
> Even in the presence of variance due to stochasticity induced by MCMC and gradient procedures, the proper tuning of the hyperparameters makes the training procedure to be successful. To practically demonstrate it, we provide the behaviour of $\mathcal{L}_2$-UVP metric (between the unregularized ground truth barycenter mapping and our learned entropic barycenter mapping $\pi^{f_k}(y \vert x_k)$) for our Ave Celeba experiment, see Figure 3 in the attached pdf. We emphasize that $\mathcal{L}_2$-UVP directly measures (by computing pointwise MSE) how the learned mapping differs from the true mapping. So, the results showcase that the presence of stochasticity in the loss does not invalidate the optimization.
>
> **(Q1.1) Benchmarking the performance**
>
> In order to benchmark the performance, one either needs to know the true barycenter (it holds true for the majority of experiments we consider) or utilize some problem-specific metrics to check whether the recovered barycenter satisfies some expected problem-specific properties. The particular loss curves during the training are indeed not that informative.
>
> **Remark 1. "ULA is surely not the simplest MCMC algorithm."**
>
> We agree that we were a little hasty with this statement. Will be fixed in the revised version. Thank you!
>
> **Remark 2. "[...] the notation feels a bit heavyhanded and can be distracting from the main story of the paper."**
>
> We will simplify the background section in the final version of our manuscript by removing/moving to the appendix some not vital definitions/results. Please see our first answer to reviewer **unyK**.
>
> **Concluding remarks.** Kindly let us know if the clarifications provided address your concerns about our work. We're happy to discuss any remaining issues during the discussion phase. If you find our responses satisfactory, we would appreciate it if you could consider raising your score.

---

> > ### Comment · Reviewer_xS43 · 2024-08-08
> > **Response to Rebuttal**
> >
> > Thanks for clarifying the nuance to Theorem 4.6. This point is quite subtle and it may be worth making a statement explicitly in the text that this more than just a result of universal approximation.
> >
> > With regards to remarks about NP-hardness -- I'm not asking you to comment on this, I'm saying that in general it is hard to measure if you've succeeded in solving your barycenter problem. This is a general remark about the circumstances where you'd actually like to discover a barycenter using your algorithm, and contextualizing the experiments done in that regard.
> >
> > While it is interesting and good to see that proper hyperparameter tuning can make the training procedure successful, the reviewer definitely thinks the community reading the paper, who may want to further research it, understand some of these challenges. Indeed, more practical means of reducing the variance of the loss b/c of this reliance on MCMC would definitely be interesting, and contextualizing if this is a real challenge would be quite valuable :).
> >
> > Thanks for your detailed response. I will raise my score a point!

---

> > > ### Comment · Reviewer_xS43 · 2024-08-08
> > > **De-anonymization**
> > >
> > > One final remark -- going forward, please don't reference the arxiv version of your paper, which totally de-anonymizes you. You did this in the response to reviewer  unyK. While I understand they were asking you to compare your method to another paper, and it may be hard to do that without explicit referencing, it's important to not break the "double blind" reviewing criteria.

---

### Author Rebuttal · Authors · 2024-08-07

Dear reviewers, thank you for taking the time to review our paper! It is a great pleasure for us that you positively evaluated our paper, emphasizing the importance of the considered problem, our solid theoretical contribution and well-tailored experimental validation. Please find the answers to your shared questions below and answers to your non-intersecting questions in the corresponding replies to your reviews.

**(1) Our work vs. Energy-guided EOT paper [1] ([77] from the original manuscript). (reviewers unyK and LoLb)**

Following your request, we explain the theoretical, methodological and practical efforts needed to develop the ideas from [1] and, eventually, come up with our work.

In [1], the authors solve the Entropic *Optimal Transport* (EOT) problem, while here, we built atop their methodology to solve the Entropic *Barycenter problem*. These two problems are linked through their use of optimal transport tools, but they are otherwise unrelated.  In particular, the theoretical guarantees for the EOT problem do not imply guarantees for the entropic barycenter problem, and the latter requires theoretical guarantees, built on distinct tools.  Our paper fills the theoretical and implication gaps for the entropic barycenter problem.

We would like to begin by underscoring the **novelty** of our key theoretical result, namely Theorem 4.1, whose derivation was highly non-trivial. The beauty of our Theorem is that it allows solving the barycenter problem without explicit knowledge of the barycenter distribution itself. This *drastically differs* from the classical EOT problem considered in [1], where both marginal distributions are known through their samples.  Unlike the EOT problem, the nature barycenter problem studied in our paper requires novel technical insights, such as being able to accommodate multiple potentials and observing that these necessarily have to obey a certain congruence condition, see Theorem 4.1 and line 167. These, and several other technical obstructions, imply that there is a lengthy mathematical and technical leap from the theoretical guarantees/implementation peculiarities in [1] to those which we derived in our manuscript.

In addition, our work is not limited by the proposed methodology for solving the entropic barycenter problem.  The technology, both theoretical and code, which we developed for solving the entropic barycenter problem, has a number of non-trivial future implications for the solvability of other problems in the broader area of computational optimal transport.  For instance, we highlight our concept with learning barycenter on an image manifold of a pre-trained StyleGAN; which allows for *meaningful* recovery of the barycenter distribution $\mathbb{Q}^*$ and would lead to new applications of our technology (see Section B.2 in the Appendix). Secondly, we have pioneering statistical learning theoretic results for the Entropic barycenter problem in (Section 4.3.), which may be of independent interest in the learning theory sub-community. Specifically, our Theorem 4.5 shows that the representativeness $\mathbb{E} Rep_{X_k}(\mathcal{F}_k^{C_k}, \mathbb{P}_k)$ with respect to the class of entropic $C_k$-transforms can be bounded by a rate independent of the functional class $\mathcal{F}$ and the cost $c_k$ functions' properties, which seems to be a bit counterintuitively.

**(2) New experiment with the biological data (reviewers unyK, wGnY)**

Following the suggestion of reviewer unyK and the request of reviewer wGnY to consider more practical tasks, we considered the problem setup of predicting the interpolation between single cell populations at multiple timepoints from [3].  The objective is to *interpolate* the distribution of cell population at any intermediate point in time, call it $t_i$, the cell population distributions at past and future time-points $t_{i + 1}$ and $t_{i - 1}$.  Since this is an interpolation problem, it is natural to expect that the intermediate population is a (entropy-regularized) barycentric average (with $\ell^{2}$ cost) of both the population distributions available at the nearest preceding and future times. We leverage the data pre-processing and metrics from paper [2], wherein the authors provide a complete notebook with the code to launch the setup similar to [3].  There are 3 experimental subsetups with dimensions $D=50,100,1000$, and the metric is MMD;  see [3] or [2, Section 5.3] for additional experimental details. We report the result in **Table 1 in the attached PDF** where we find that in most cases, our general-purpose entropic barycenter approach nearly matches the performance of the leading computational in computational biology, which are specifically designed to solve only this problem. This underscores the scope of problems in which our barycentric optimal transport technology can act as a viable foundation model, directly out-of-the-box.

**PDF content**.

We attach **PDF** file that contains additional experiments for the rebuttal. A brief explanation of the content:

* We provide evidence of our method's robustness for different entropic regularization setups on the MNIST 0/1 dataset (**wGnY**). This experiment demonstrates that the smaller the regularization term $\epsilon$, the better preserving the image content.
* We demonstrate convergence $\mathcal{L}$2-UVP of our method during the training phase on the Ave, Celeba! dataset (**kw6j, xs43**).
* We show the performance of our method depending on the number of Langevin steps in the case of Gaussian distributions' barycenter (**wGnY**).
* We provide the practical application of our method with Single-cell experiment, comparing our approach with existing solvers (**wGnY, unyK**).

**References**

[1] Mokrov et. al., Energy-guided entropic neural optimal transport. ICLR'2024.

[2] Korotin et. al., Light Schrödinger Bridge. ICLR'2023

[3] Tong et. al. Simulation-free schrödinger bridges via score and flow matching. AISTATS'2024

---

### Decision · Program_Chairs · 2024-09-25

**Decision:**

Accept (spotlight)

**Comment:**

The paper proposes a new methodology for estimating entropic regularized Wasserstein barycenters of $K$ distributions, for arbitrary transport cost function $c$. This approach is based on a dual formulation of the original problem that the authors establish, leveraging the weak entropic $c$-transform. This dual problem consists in maximizing a convex combination of functionals, over $K$ continuous functions under some constraints. Based on this result, the authors provide an algorithm to solve this dual problem, using stochastic gradient descent and MCMC samplers. The proposed approach is well contextualized in relation to existing work and comes with both theoretical results and interesting numerical illustrations with real-world data.

All the reviewers and I found that the paper introduces interesting and novel ideas for solving the problem at hand, which are well-developed. Therefore, we think that the paper is worth accepting.